# LEARNING IS FORGETTING:
# LLM TRAINING AS LOSSY COMPRESSION

**Henry C. Conklin**[○,•]**, Tom Hosking**[•]**, Tan Yi-Chern**[•]**, Sarah-Jane Leslie**[○]**, Jonathan D. Cohen**[○]**,
**Thomas L. Griffiths**[○]**, Max Bartolo**[•]**, Seraphina Goldfarb-Tarrant**[•]
[○]Princeton University
[•]Cohere
`henry.conklin@princeton.edu,`

## ABSTRACT

Despite the increasing prevalence of large language models (LLMs), we still have a limited understanding of how their representational spaces are structured. This limits our ability to interpret how and what they learn or relate them to learning in humans. We argue LLMs are best seen as an instance of lossy compression, where over training they learn by retaining only information in their training data relevant to their objective(s). We show pre-training results in models that are optimally compressed for next-sequence prediction, approaching the Information Bottleneck bound on compression. Across an array of open weights models, each compresses differently, likely due to differences in the data and training recipes used. However even across different families of LLMs the optimality of a model's compression, and the information present in it, can predict downstream performance on across a wide array of benchmarks, letting us directly link representational structure to actionable insights about model performance. In the general case the work presented here offers a unified Information-Theoretic framing for how these models learn that is deployable at scale.

## 1 INTRODUCTION

We still have a limited understanding of *how* Large Language Models (LLMs) achieve impressive results across a wide array of tasks (Devlin et al., 2019; Grattafiori et al., 2024). While a growing body of work interprets LLMs using behavioural experiments, probing, or causal interventions, the scale of these models makes understanding how their representation spaces are structured a continued challenge. Here we look at an LLM as an instance of lossy compression, offering an account of how models represent information during training and what information matters for performance.

Lossy compression represents data efficiently by preserving only the information from a source relevant to a goal. While audio recordings intended for human listeners can be gigabytes in size, MP3 files save space by discarding frequencies typically outside the range of human hearing (Jayant et al., 1993); similarly, a JPEG file omits subtle colour variations that are difficult for the human eye to perceive. We draw a parallel with LLMs, which are expected to generate responses humans prefer, after being trained on trillions of tokens – more language data than a human hears in 200 lifetimes. More generally, compression is thought to underpin learning in both humans and models (see Feldman, 2016), giving a formal account of LLM pre-training in terms of compression allows us to work towards a unified theory of representation learning. We present results showing that over the course of pre-training LLMs optimally compress the information present in their training data for next sequence prediction.

Compression is inherently opinionated – some information from the source is preserved, some is forgotten to save space. Information Theory (Shannon, 1948) provides a formal language to describe this process, letting us both quantify the information present in a representation *and* compute a bound where it is optimally compressed with respect to the data it represents. Our results build on the Information Bottleneck (IB) theory of deep learning (Tishby & Zaslavsky, 2015), showing pre-training follows a two phase trajectory: first increasing mutual information with the training objective, before compressing input information. Across a wide array of LLMs we find each model

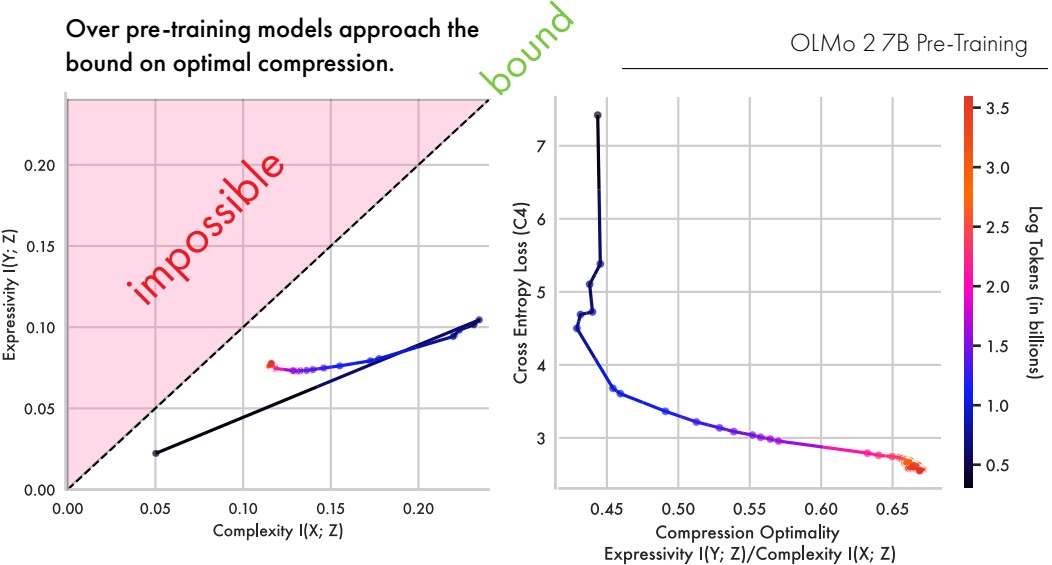

Figure 1: **LLMs Learn an Optimal Compression of the Internet (Left)** The information plane for pre-training of the OLMo2 7B model. The horizontal axis shows mutual information between representations and the input (complexity), the vertical axis shows mutual information with the predicted output (expressivity). The dotted line indicates the bound where models are optimally compressed, hue indicates timepoint in training in terms of tokens in billions. Estimates are based on 10,000 samples from the C4 dataset which is a broad crawl of the internet. **(Right)** The vertical axis shows the OLMo2 7B model's loss on next-token-prediction of C4. The Horizontal axis shows the model's proximity to the bound. Representations begin to approach the bound as the loss saturates.

compresses differently, with the optimality of a model's compression and the information it preserves predicting performance on downstream benchmarks.

A hallmark of large-scale distributed systems, like neural networks, is that they are difficult to understand as a function of their parts alone (Anderson, 1972; Mitchell, 2009). Our approach to interpretability allows us to consider learning and generalisation at the scale of an entire model, rather than studying individual circuits or neurons within it. Additionally it allows us to frame how models do so well at so much in terms of existing theories of learning and compression, while providing actionable insights at LLM scale.

In what follows we focus on offering concrete answers to three questions: Do LLMs optimally compress their representations? What information survives that compression? What representational structures drive performance? In summary, the core findings are:

- Pre-training dynamics for LLMs closely follow theoretical predictions from the Information Bottleneck, with models first expanding representations before slowly approaching optimal compression.

- Scale conditions these dynamics, with smaller models (below 7 billion parameters) struggling to achieve meaningful compression later in training.

- How optimally compressed a model is correlates significantly with performance across six benchmarks for six families of open-weights large language models, letting us directly relate representation structure to behaviour.

- By quantifying the amount of preference information in a model we get a quantification of how aligned representations are with preference distinctions which significantly predicts downstream performance across 47 LLMs ($r = 0.76, p < 0.001$).

- Finally, we compare a wide array of open-weight models across 5 model families, showing they all converge near optimal compression.

## 2 BACKGROUND & RELATED WORK

### 2.1 LEARNING, INFERENCE, AND COMPRESSION

Compression has been argued to underpin learning and inference in humans (Chater, 1997; Chater & Vitányi, 2003; Feldman, 2000; Pothos & Chater, 2001) and models (MacKay, 2003; Poggio et al., 2004). Increasingly, probabilistic inference and complexity minimisation are seen as deeply intertwined (Feldman, 2016) – a point perhaps made clearest by Bayesian inference, which implicitly prefers the simplest hypotheses consistent with observed data (Edwards, 1972; Jeffreys, 1939; Vitányi & Li, 2000). Bayesian approaches to human cognition offer accounts of how a broad array of human behaviour can be productively thought of as this kind of inference (see Griffiths et al., 2024, for a review). In machine learning Occam's Razor has long been used as a model selection criterion – where the best model is the simplest one consistent with the data (Burnham & Anderson, 2002; Rissanen, 1978; Wallace & Boulton, 1968). The bias variance trade-off (Geman et al., 1992) makes this explicit in the context of neural networks, showing more complex models may achieve better fit to the training data, but they also generalise worse than their simpler counterparts. While some work has studied whether or not LLMs can match lossless compression algorithms in-context (e.g. Delétang et al., 2023), this is distinct from giving an account of LLM training itself as a process of lossy compression – the object of study here. It is worth noting that there is not universal agreement about how to assess compression (see MacKay, 2003, for discussion), but here we follow in the information-theoretic tradition (Shannon, 1948).

### 2.2 RATE DISTORTION THEORY

Consider a function $\theta$ that encodes an input $X$ in a representation $Z$, $Z = \theta(X)$. This representation is then decoded by a function $\phi$ to produce predictions $\hat{Y}$ for an output with true label $Y$, $\hat{Y} = \phi(Z)$. Assuming that $X$ and $Y$ are not independent, if $\theta$ were to losslessly preserve all the information from the input, we would expect $\phi$ to be able to precisely recover the corresponding output, with $\hat{Y} = Y$. Rate Distortion Theory (RDT) (Shannon, 1948) instead considers the *lossy* case $\hat{Y} \neq Y$, where some amount of error in the prediction – **distortion** – is acceptable. It then becomes a question of how much information about the input – termed the **rate** – the encoder needs to preserve to achieve a given level of distortion.

**The Information Bottleneck (IB)** Tishby et al. (2000) looks at a particular case, where the **rate** is given as the mutual information between inputs and their representation $I(X; Z)$, and **distortion** as the mutual information between a representation and the corresponding target prediction $I(Y; Z)$ – the 2D space this creates is called the *information plane* (shown in Figure 1). Since $I(X; Z)$ reflects how much information about the input space is preserved it can be referred to as *complexity* (e.g. Zaslavsky et al., 2018). Likewise $I(Y; Z)$ is referred to as *accuracy* given it quantifies how much information a representation has about the target output it will be used to predict. To distinguish this quantity from behavioural accuracy (e.g., exact match on a task) we refer to it as *expressivity* – how uniquely a representation can refer to its target (in line with Kirby et al., 2015). Optimal compression within the IB occurs when an encoding $Z|X$ preserves only the information about $X$ relevant to predicting $Y$, or when $Z|X$ minimises

$$\mathcal{F}_\beta[p(Z|X)] = I(X; Z) - \beta I(Y; Z) \tag{1}$$

where $\beta$ is a trade-off parameter controlling the allowable level of distortion. When $\beta$ approaches 0 all inputs are compressed to a single point, as $\beta \to \infty$ we approach the lossless case, where using $X$ or its encoding $Z$ tells us the same information about $Y$; $I(Y; Z) = I(Y; X)$. The curve traced by varying $\beta$ draws a bound, where the encoding $p(Z|X)$ is optimally compressed – everything above the curve is unachievable and everything below it is suboptimal. This bound starts off with a linear relationship where $I(X; Z) = I(Z; Y)$, until $Z$ captures all information shared between $X$ and $Y$. Intuitively, in an optimal encoding each additional bit of complexity gets you an additional bit of accuracy, until all information shared by X and Y are represented; $I(Y; Z) = I(Y; X)$ – in the cases studied here all models stay well below this saturation point, so for clarity we refer to the bound as the line $I(X; Z) = I(Z; Y)$ (for further discussion of the bound and computing it numerically see Appendix E.8).

**Applying the Information Bottleneck to Deep Learning**   Tishby and Zaslavsky (2015) offered a theoretical characterisation of training a multi-layered neural network as optimising an Information Bottleneck. They theorise two phases of training: first a fitting phase during which representations increase mutual information with the target labels $I(Y; Z)$; and second a compression phase, during which models compress irrelevant information about the input $I(X; Z)$ and in the process begin to approach the optimal bound. It is this latter phase that is hypothesised to result in representations that generalise robustly.

Shwartz-Ziv and Tishby (2017) confirmed the two-phase prediction from the IB theory of deep-learning empirically in feed forward networks trained on MNIST. Subsequent work has questioned the generality of these findings, showing how – at least in linear networks – the compression phase can be driven by the type of non-linearity used (Saxe, Bansal, et al., 2019), or that compression is not necessarily required for generalisation (Goldfeld et al., 2019).It remains unclear whether deep-learning models in the general case can be expected to follow the phases of expansion and compression predicted by the IB, in particular when it comes to sequence models (e.g. Transformers) trained on complex tasks.

## 2.3   Interpreting Neural Networks

A broad literature on the theory of deep learning tries to give an accounting of learning dynamics in small multi-layer networks (e.g. Frankle & Carbin, 2018; Saxe, McClelland, & Ganguli, 2019). While there has been some extension of these kinds of representational analyses to larger models – like applying information theoretic methods to transformers (Voita et al., 2019) – much of the work on interpretability in LLMs leverages behavioural or probing evidence. Behavioural approaches treat models as akin to psycholinguistic subjects (Futrell et al., 2018, 2019), taking model outputs as behaviours (Hu et al., 2020; Marvin & Linzen, 2018; Warstadt et al., 2019). Probing (Pimentel et al., 2020; Veldhoen et al., 2016; Voita & Titov, 2020) trains a smaller model – like a linear classifier – to predict labels from a model's latent representations, as evidence that information relevant to those labels is present. While valuable, these approaches are removed from the models' representations themselves – characterising downstream behaviours rather than characterising the representational structures that drive them.

Mechanistic interpretability follows in a similar vein but aims to describe how circuits within a model implement the functions that solve a task. These analyses have given accounts of how two layer linear and non-linear models represent features from synthetic data (Elhage et al., 2021) or how single-layer attention only transformers solve modular addition (Nanda et al., 2023). When deployed at scale, to LLMs, this work often relies on training unsupervised probes termed *sparse auto-encoders* (Elhage et al., 2022) to identify correspondences between parameters and different words or concepts from the training data (Bricken et al., 2023). In the general case this work often looks for 'mono-semanticity' – looking for lossless, one-to-one correspondences between input features and parts of a model. More recently studies of when features emerge during pre-training have aligned with the expansion/compression pattern described by the IB theory (Ge et al., 2025).

To be sure, there is an abundance of methods for analysing deep-learning models. Here, however, we highlight a disconnect between work on the theory of learning in humans and neural networks, and work on interpretability. Interpretability methods can be deployed at scale on complex models and tasks, but lack clear relationship to existing theoretical work. In the sections that follow we operationalise Rate Distortion Theory, and related work on learning as compression, at LLM scale. This allows us to analyse training dynamics while contextualising our conclusions in existent and well-studied theoretical frameworks. Our approach represents one that is theoretically motivated but can be applied to any model at any scale.

## 3   Methods

### 3.1   Entropy Estimation

Let $T \in \mathbb{Z}^{B \times S}$ be a batch of $B$ tokenized samples with sequence length $S$, drawn from a corpus of text data $\mathcal{T}$, and let $\theta$ be a model with $L$ layers and representation dimension $h$; the corresponding encoded representations are $Z \in \mathbb{R}^{L \times B \times S \times h}$. Let $X \in \mathbb{Z}^{B \times S}$ be feature labels for the text in $T$. For example, when we look at optimal compression with respect to the IB bound, these labels $X$

are the token ids for the model inputs; however, when analysing representation information more generally, these can be other input features, such as preference label or language id. It is desirable to compute the mutual information $I(X; Z)$ using Shannon entropy as opposed to differential entropy to accomplish this, previous work quantises $Z$ into $n$ bins, to get a discrete encoding $\hat{Z}$ (Shwartz-Ziv & Tishby, 2017; Voita et al., 2019)[1]. Unfortunately the approaches from this previous work have memory and resource requirements that make them difficult to apply at LLM scale. As a result we use the soft-entropy estimator from Conklin (2025) – this is an efficient differentiable relaxation of a binning-based estimate that has been shown to converge to the true entropy of a distribution. We describe the estimation process in detail below, this estimator is not original to our work but we are the first to apply it to analyse LLMs using rate distortion theory.

To obtain a soft quantisation $\hat{Z}$, this approach first computes $\bar{Z}$, which is the normalization of $Z$ to lie on the surface of the unit sphere $\mathbb{S}^h$ in $\mathbb{R}^h$. It then samples $n$ points $\{w_i\}_{i=1}^n$ uniformly at random from $\mathbb{S}^h$.[2] Then, for each normalized representation $\bar{z} \in \mathbb{R}^h$, we compute a vector whose $i^{th}$ entry is the cosine between $\bar{z}$ and $w_i$, then apply softmax to that vector – softly assigning each embedding $\bar{z}$ to the points in $W$. More formally, for each $(l, b, s) \in [L] \times [B] \times [S]$, tensor $\bar{Z}$ (whose shape coincides with $Z$) is defined so that $\bar{Z}_{l,b,s,:} = Z_{l,b,s,:}/\|Z_{l,b,s,:}\|$, and we stack the uniform samples $\{w_i\}_{i=1}^n$ into a matrix $W \in \mathbb{R}^{h \times n}$. Tensor $\check{Z} \in \mathbb{R}^{L \times B \times S \times n}$ is then defined so that for $(l, b, s) \in [L] \times [B] \times [S]$,

$$\{w_i\}_{i=1}^n \sim \text{Unif}(\mathbb{S}^h), \qquad W_{:,i} = w_i, \qquad \check{Z}_{l,b,s,:} = \text{softmax}\Big(\frac{\sum_{j=1}^h \bar{Z}_{l,b,s,j} W_{j,:}}{\epsilon}\Big). \qquad (2)$$

Where $\epsilon$ is a temperature parameter, which we set to enable direct comparison of representations with different dimensionalities following the procedure described in Appendix E.1.1. Each vector $\check{Z}_{l,b,s,:}$ defined this way is a probability vector. Let $\hat{Z} \in \mathbb{R}^{L \times n}$ be the matrix obtained from tensor $\check{Z}$ by averaging over the batch and sequence dimensions, and let $\hat{z}_l$ be the $l$-th row of this matrix, a probability vector of length $n$ by construction:

$$\hat{Z} = \frac{1}{BS} \sum_{b=1}^B \sum_{s=1}^S \check{Z}_{:,b,s,:}, \qquad \hat{z}_l = \hat{Z}_{l,:}, \quad H(\hat{z}_l) = -\sum_{j=1}^n \hat{z}_{l,j} \log \hat{z}_{l,j}. \qquad (3)$$

Vectors $\hat{z}_\ell$ are probability vectors for each layer $l \in [L]$ describing a categorical distribution over $n$ categories. Therefore we can compute the Shannon entropy $H(\hat{z}_l)$ as above. Due to the normalisation step during quantisation, this distribution approximates the probability that a representation in a layer $l$ lies along a particular angle with respect to the origin. To estimate the entropy in an entire model, denoted $H(Z)$ we average entropy across layers. Efficiency (Wilcox, 1967) normalises $H$ by the entropy of a uniform distribution $\log(n)$, thereby bounding the entropic quantity between 0 and 1 – to aid interpretability here we convert $H(Z)$ to an efficiency $\mathcal{H}(Z)$ by additionally normalising by the entropy of a uniform distribution at each layer. These definitions can also be conditioned on the feature labels $X$.

$$\mathcal{H}(Z) := \frac{1}{L \log(n)} \sum_{l=1}^L H(\hat{z}_l), \qquad \mathcal{H}(Z|X=x) := \frac{1}{L \log(n)} \sum_{l=1}^L H(\hat{z}_l|X=x) \qquad (4)$$

This now allows us to efficiently compute the mutual information between input features $X$ and encodings across an entire model, regardless of model size.

$$I(X; Z) := \mathcal{H}(Z) - \sum_{x \in X} P(X=x)\mathcal{H}(Z|X=x) \qquad (5)$$

## 3.2 MUTUAL INFORMATIONS & BACK-OFF

To look at whether or not a model is optimally compressed with respect to some data we need to compute mutual informations with respect to input and output labels. LLMs are trained with inputs as preceding context and outputs as trailing context (for discussion of this, and examples of

---

[1]For discussion of Shannon entropy and why previous approaches are not scalable see Appendices E.6,E.7.

[2]This is equivalent to sampling from an isometric $h$-dimensional multivariate normal, $\tilde{w}_i \sim \mathcal{N}(0, Id_h)$, and scaling to unit length, $w_i = \frac{\tilde{w}_i}{||\tilde{w}_i||}$.

the labelling procedure see Appendix E.4). Maintaining conditional estimates of a token embedding given a preceding context $P(Z|X)$ for every possible context window proves intractable, and many contexts occur only once in the training data. Accordingly, like many other works on language modelling we approximate the distribution over possible sequences using n-grams with a kind of back-off (Katz, 1987). By conditioning on finite widths of preceding context we can tractably approximate $P(Z|X)$; the maximum width we consider here are quad-grams by which point $I(X;Z)$ begins to converge and past which point computation becomes intractable in an LLM setting. By backing off further (e.g. to trigrams, bigrams, and tokens) we can also estimate how much different context widths contribute to information in a model - for clarity the majority of results use token level backoff, with other levels of backoff noted where they're presented. We vary the degree of backoff equally for both the input $P(Z|X)$ and output $P(Z|Y)$ distributions, this is because during training a model receives gradient information from the full trailing context $Y$ due to teacher forcing (see Appendices E.2, E.3 for further discussion and visualisation of the labelling procedure). In comparing different models we would like to be able to determine how close a given representation system is to the IB bound – by extension, how optimally compressed it is. When on the bound, representations preserve only the information from the input relevant to predicting the output. We quantify this with a summary statistic *optimality*.

$$\text{Optimality} = \frac{\text{Expressivity}}{\text{Complexity}} = \frac{I(Y;Z)}{I(X;Z)} \tag{6}$$

Intuitively this quantity approaches 1.0 as a representation system approaches the bound, regardless of where along the bound the system is placed (i.e. which beta value along the bound the system is closest to). More generally this is a relative quantity reflecting how many bits of expressivity a system has for each bit of complexity.

In addition to mutual information with input and output labels, we also consider preference data. A growing body of work stresses the importance of post-training approaches for aligning models with human preference (Bai et al., 2022; Ouyang et al., 2022; Rafailov et al., 2023). We can quantify this information in a model using preference data, where a prompt has two continuations, one of which is labelled preferred by raters the other labelled as rejected. Conditioning on this label lets us compute $P(Z|\text{preferred})$ and $I(Z;\text{preferred})$.

**Data and Sampling** Getting a true estimate of the entropy of a vector space remains a major challenge, with most approaches underestimating the true entropy (Paninski, 2003). As a result we do not claim our experiments estimate the entropy of a model's true latent distribution, but rather an estimate of the entropy with respect to a particular sample of data. By holding the data constant across models and experiments we can compute an estimate that is useful for comparisons, even if it does not exactly match the true entropy. Unless otherwise noted, token bigram, trigram, and quad-gram estimates are with respect to 10,000 samples from C4 (Raffel et al., 2020), and preference estimates are based on 10,000 samples from Tulu (Lambert et al., 2024); in both cases we consider a maximum context length of 512.

## 4 EXPERIMENTS

In order to study training time-courses our pre-training analyses look at the OLMo2 family of models (OLMo et al., 2025), which makes available intermediate checkpoints [3]. We focus analysis on the 7B model unless otherwise noted, while including results for the 32B and 1B variants to show where conclusions hold or differ across model scales. In addition, to show our conclusions hold outside of this particular family of models we compare a wide array of open-weights LLMs (which do not make intermediate training checkpoints available), showing where they lie on the information plane at the end of training.

---

[3]Appendix D includes additional pre-training analyses of the Smol LM2 (Allal et al., 2025) and Pythia (Biderman et al., 2023) models which also make intermediate checkpoints available. These follow a similar pattern to the results presented here.

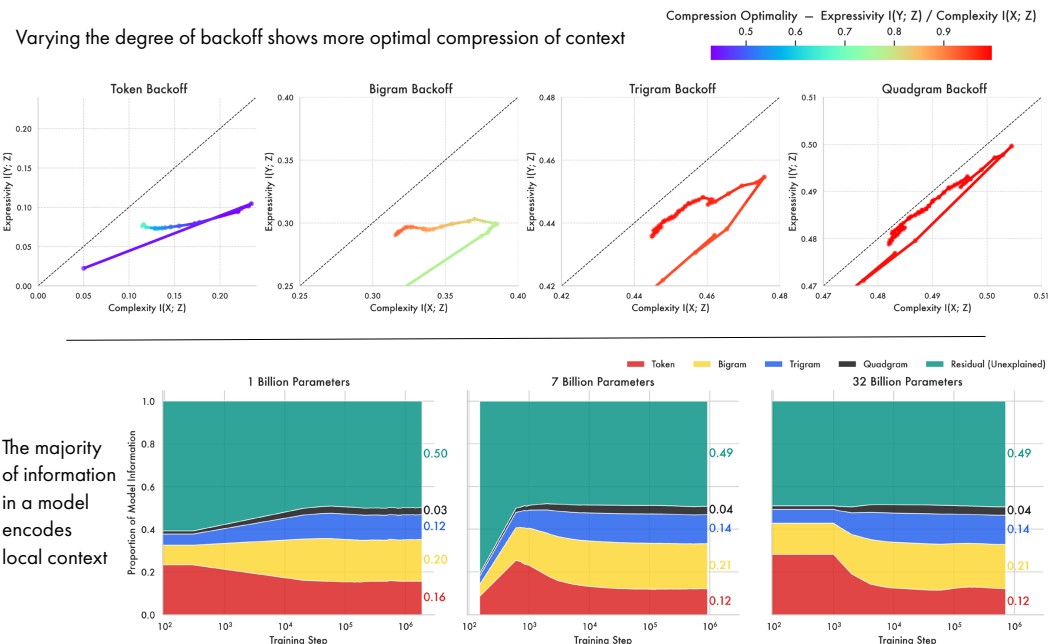

Figure 2: **Models Largely Encode Local Context**. (Top) The information plane over pre-training for the different levels of backoff. By changing how many tokens we condition the mutual information on in the context window, we see how the OLMo2 7B model compresses not just token but also local context information. Across all context windows we see the same two phase pattern predicted by the Information Bottleneck – with more contextual representations approaching greater optimality - indicated by hue. As context increases models compress both the target and source over training, rather than compressing the target independently. We hypothesise this is because in language modelling with full context the target and source distributions are nearly identical. (Bottom) By computing the conditional mutual information for a level of back-off given the others we can quantify what proportion of a model's information encodes each level of context information. Each facet shows a different model size, with the horizontal axis reflecting training step and the vertical axis reflecting proportion of information – hue indicates level of back-off

## 4.1 PRE-TRAINING APPROACHES OPTIMAL COMPRESSION

The majority of pre-training appears to be a slow compression of a model's training data. The Information Bottleneck theory of deep learning predicts two phases: a *fitting phase* during which output information $I(Y; Z)$ increases, followed by a *compression phase* during which input information $I(X; Z)$ decreases and representations approach the bound. This transition to compression is believed to occur when error on the training set saturates. Shown in figure 1 (and reproduced in Figure 2) is the training trajectory for the OLMo2 7B model with respect to data from English C4. Strikingly, the 7B model closely follows the two-phase prediction from the Information Bottleneck, first increasing mutual information with outputs, before compressing input information and progressing towards the bound on optimal compression. Additionally this transition appears to happen as the model's loss on next-token prediction begins to saturate (figure 1 right). This shows how, even at scale, deep-learning models appear to thread a needle between representational complexity and expressivity. It also demonstrates how LLMs can be effectively studied from the perspective of Rate Distortion Theory, as they try to converge to an optimal lossy compression of their training data.

**Models More Optimally Compress Contextual Information** By varying the degree of back-off in the source and target distributions used to compute mutual information, we can see how contextual information evolves over pre-training at the token, bigram, trigram, and quad-gram levels (Figure 2 top). All cases result in a similar two-phase pattern of expansion and compression, with larger conditioning context converging closer to the bound (indicated by hue). For token-level back-off late training aligns with previous work on MNIST, with models compressing the source distribution

– reducing complexity – while maintaining expressivity. At higher levels of contextualisation both complexity and expressivity are reduced. We hypothesise this is because in language modelling the source and target are sampled from the same distribution, what counts as an 'input' vs. an 'output' is a product of what point in the sequence the model is during generation unlike cases with an inherent difference between the distributions (e.g. MNIST classification where the source is over images and the target is over discrete labels). While individual tokens, or words can have divergent source and target distributions – an adjective almost always precedes a noun and follows a determiner; permuting that order will render a sentence ungrammatical – at the phrase, sentence, or paragraph level the difference between preceding and trailing context – between target and source – becomes harder to identify. Making it difficult to compress one without also compressing the other, resulting in the reduction of both complexity and expressivity in the trigram and quadgram facets of figure 2. The higher degree of optimality in contextual encodings likely reflects an inherent pressure in the pre-training objective for models to not only develop token representations, but representations of a token *in context*.

**Embeddings Largely Encode Local Context**  We compute the proportion of information in a model explained by each level of back-off in the source distribution independently (a detailed explanation of this process with results related to performance can be found in Appendix E.5). As shown in figure 2 (bottom) the majority of information in a model encodes local context (token to quadgram), likely reflecting the information locality of the natural language on which they're trained (Gibson, 1998; Gibson et al., 2000; Hahn et al., 2022). The 1 billion parameter model also has more token information and less contextual information than its larger counterparts. The residual information likely encodes the finer-grained contextual distinctions found in the remainder of the 512 token context window (i.e. information up to an n-gram width of 512) – given the sparsity of n-grams greater than a quadgram those mutual informations are intractable for us to compute.

**The Effect of Scale: Smaller Models Struggle to Compress**  Parameter count shows a marked effect on the degree of compression achievable by a model. Figure 3, shows pre-training trajectories for the 1B, 7B, and 32B parameter models. The larger models both closely follow the hypothesized Information Bottleneck trajectory, exhibiting phases of expansion and compression, ultimately approaching optimal compression. The 1B parameter model exhibits markedly different behaviour. While it successfully completes the initial expansion phase – increasing output information $I(Y;Z)$ – it fails to approach optimal compression. Instead, in the second phase the smaller model oscillates while moving slowly away from the bound (Figure 3 bottom-left). This suggests that for a given level of data complexity, a certain parameter threshold may be necessary for models to achieve an optimal compression – an observation in line with work on scaling laws (Kaplan et al., 2020).

**A Wide Array of Open-Weight Models Converge Along the Bound**  In addition to looking at the OLMo2 family of models, we compute complexity and expressivity estimates across a diverse array of open-weight models. A striking convergence pattern emerges: across different model families, hyper-parameters, and training methodologies, representations ultimately converge to representations clustered near the bound on compression (Figure 3, *Top Left*; with full model names in Appendix, Figures 5 and 6). More than that models all approach the same point on the bound, suggesting they all converge to a similar information structure. This suggests that training as a process of compression is not an artifact of a single LLM's training trajectory, but more fundamentally applies to deep-learning models as a class, and to the data and the objectives used to train them.

## 4.2 Relating Representation Structure to Performance

So far we have studied how information in an LLM is structured; we now consider how that structure relates to downstream performance. Figure 4 shows correlations between representational measures and aggregate performance across six benchmarks (MMLU Pro, BBH, Math LVL5, IFEval, GPQA, MuSR) for 47 open weights models from 6 different families (Evaluations and aggregation from Fourrier et al., 2024). With token back-off lower Complexity relates significantly to performance ($r = -0.38$, $p = 0.006$), while expressivity alone does not ($r = 0.08$, $p = 0.575$) - Figure 4 (top). However, the ratio between expressivity and complexity, our measure of how close a model is to optimal compression, is a significant predictor of downstream performance($r = 0.52$, $p < 0.001$).

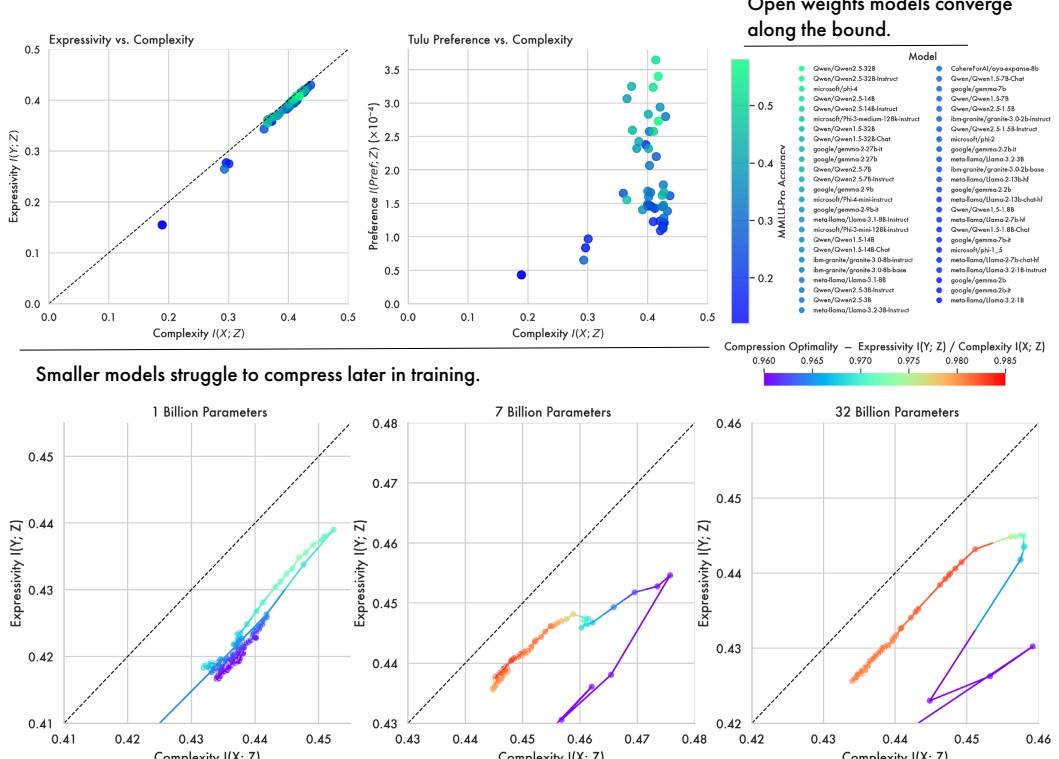

Figure 3: **Models Converge Along the Bound With Smaller Models Struggling to Compress**. **(Top Left)** Open-Weights models across 6 families at the end of training, lie along the bound on optimal compression. Hue indicates performance on MMLU Pro. **(Top Right)** The vertical axis indicates mutual information with preference, with models with more preference information exhibiting better performance **(Bottom)** Zooming in on later pre-training for each model size the 1B model matches Phase 1 but struggles to achieve meaningful compression later on, oscillating for much of pre-training off the frontier. All results use back-off to the trigram level. A full legend identifying each dot, with additional levels of back-off, is given in Appendix Figures 5 and 6.

While LLMs approach optimal compression for next sequence prediction over pre-training, a large body of work also tries to improve their ability to follow instructions, and generate responses humans prefer (e.g. Ouyang et al., 2022). We use preference data (Lambert et al., 2024) to compute mutual information with preference. As shown in Figure 4 (bottom right), the amount of preference information in a model proves a significant predictor of downstream performance ($r = 0.76$, $p = 0.001$). This suggests that not only does the optimality of a model's compression matter, but exactly what information survives that compression does too. In Appendix C we include results showing that post-training can increase the amount of preference information across different models in the Llama family while minimally changing their complexity. This suggests that pre-training is responsible for the broad compression learned by a model, while post-training edits the information it contains; we leave a more complete assessment of how different phases of training affect representational compression to future work.

These results also indicate how the information theoretic approach taken here could potentially be leveraged during training. Two applications could be as a stopping-criterion – ceasing pre-training when distance to the bound no longer decreases, or as a model-selection criterion – picking the checkpoint that is the most optimally compressed, or with the highest proportion of preference information. Given the estimates here are computed with a single-forward pass using teacher forcing, computing an entropy estimate for candidate selection would be substantively less costly than evaluating a model across a suite of benchmarks. We look to experimentally validate these potential use cases of our approach in future work.

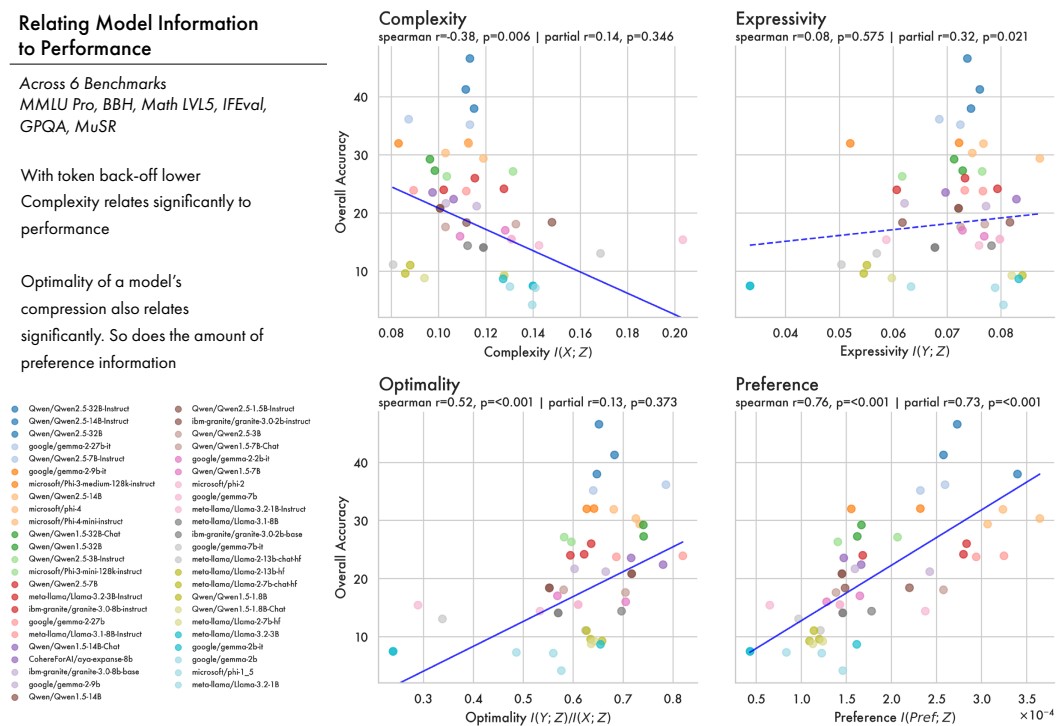

Figure 4: **Representation Information Relates Significantly to Performance** Vertical axes, shared across all plots, show aggregate performance across 6 benchmarks (MMLU Pro, BBH, Math LVL5, IFEval, GPQA, MuSR). Horizontal axes use token back-off to show complexity significantly correlates with downstream performance **(Top Left)**, while expressivity alone does not **(Top Right)**. The ratio of how many bits of expressivity a model has per bit of complexity does correlate significantly – this quantity indicates how optimally compressed a model is **(Bottom Left)**. The amount of preference information in a model also correlates with downstream performance **(Bottom Right)**. Above each facet are results from a spearman correlation between their axes, and a partial spearman that treats a model's number of parameters as a covariate. Expressivity and complexity are calculated using token back-off in all four facets.

## 5 CONCLUSION

The work presented here bridges the gap between theoretical accounts of learning and the practical complexities of LLMs. We show that LLMs learn an optimal compression of the data on which they are trained, with a wide array of open-weights models converging along the IB bound – with the optimality of a model's compression predicting downstream performance. Each compression is different; we can account for the information that survives the compressive process, showing how representations encode information about different levels of local context and human preferences.

The approach to interpretability we introduce here interprets a model as a whole – rather than focussing on a particular circuit, or attention head – because complex distributed systems are not best understood in terms of their parts alone. Giving a holistic account of what it means to train an entire model on the entire internet is a challenge, but we argue that LLMs are best understood as lossy compression. In doing so, we place them in the context of a long history of work on representation learning across the sciences.

## 6 ACKNOWLEDGEMENTS

We would like to thank Kenny Smith for his role in developing the core ideas presented here in earlier versions of this project. We also thank Julian Gold for discussion, help with formalisation, and for contributing the temperature calibration.

## 7 ETHICS STATEMENT

All experiments reported here use publicly available datasets and pretrained models obtained under their original licenses; see Appendix B.1 for details. To our knowledge, these datasets contain no personally identifiable information, and we are in compliance with their terms of use. No additional data were collected. More generally, all authors have read and adhered to the ICLR Code of Ethics. To the best of our knowledge, these results and their dissemination do not raise any ethical concerns.

## 8 REPRODUCIBILITY STATEMENT

All datasets and pre-trained models used in our experiments are publicly available (see Appendix B.1). All code will be released with an open source license. Appendix B.2 contains details of compute resources necessary to reproduce these findings.

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

## A    OPEN-WEIGHTS MODELS, DETAILED VISUALS

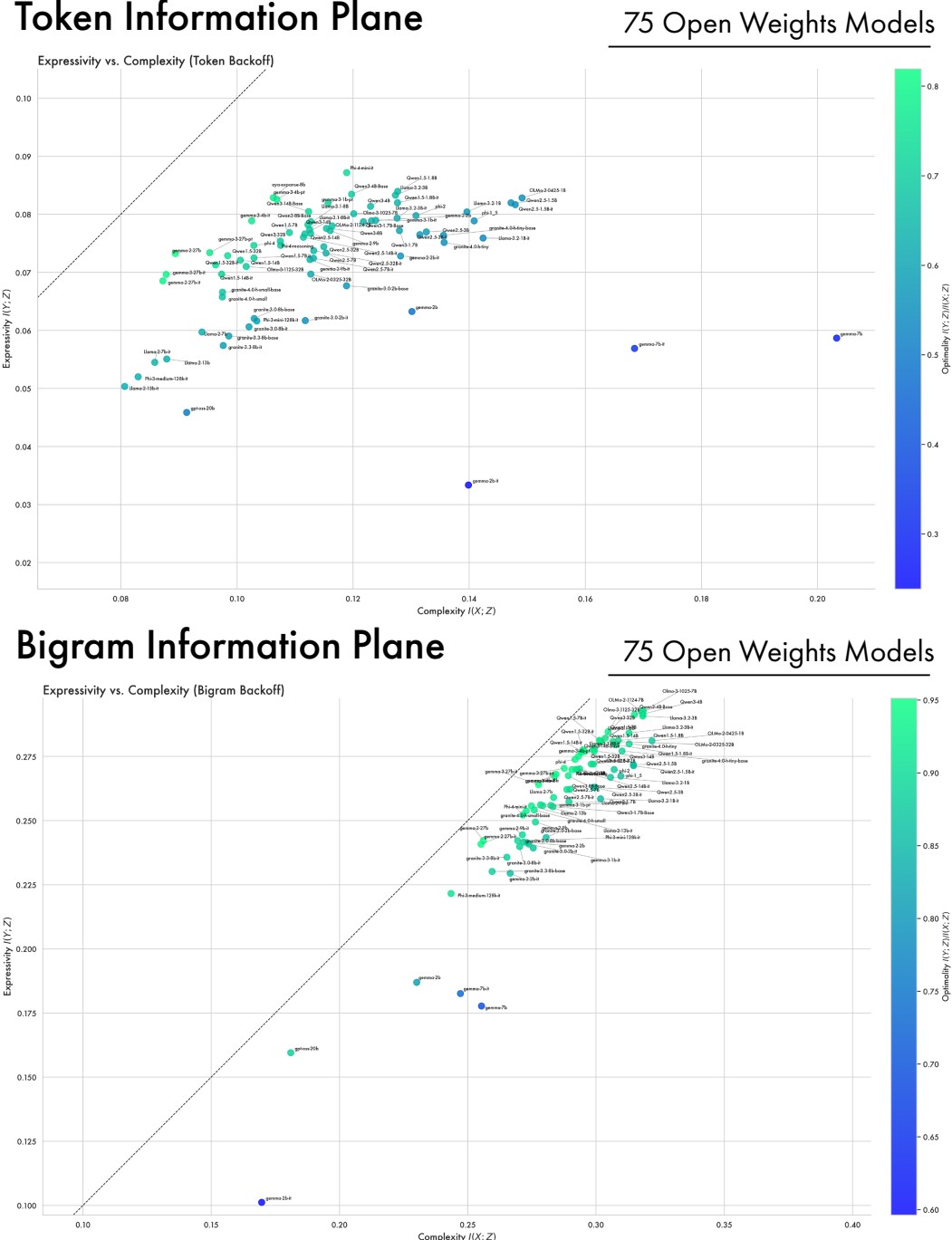

Figure 5: **Token & Bigram Information Plane for Open Weights Models** Shown here is the full, labelled token information plane for 75 different open-weights models. Overall while model lie at different levels of complexity and expressivity they broadly approach the IB Bound on optimal compression, hue indicates optimality - or proximity to the bound.

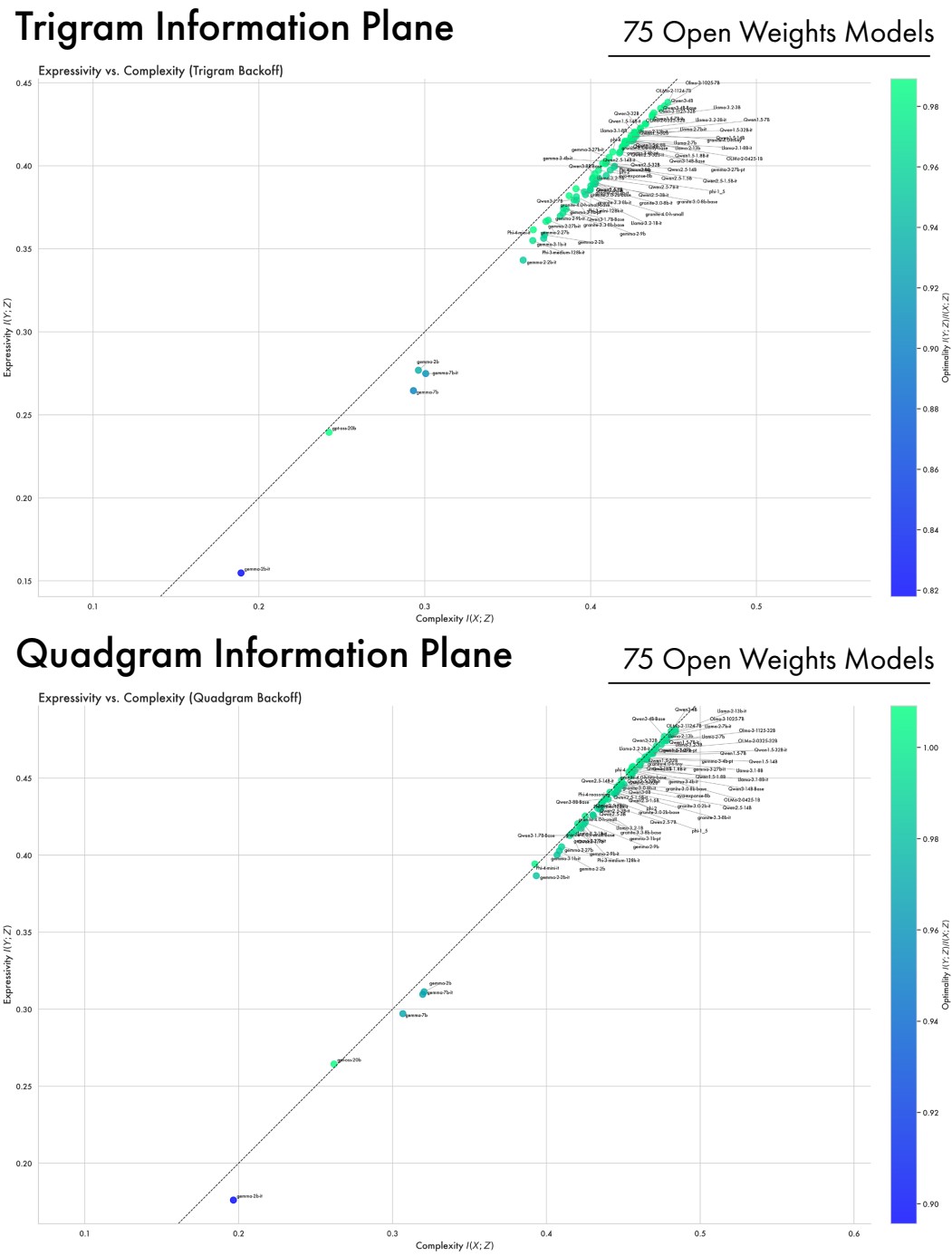

Figure 6: **Trigram and Quadgram Information Plane** Shown here is the full, labelled trigram and quadgram information plane for 75 different open-weights models. Compared with the token case above, here models lie even closer to the frontier. The quadgram estimates are noisy due to sample sparsity, this combined with the fact that all models are close to the bound results in some estimates appearing to cross the bound.

## B  DATASETS, MODELS, AND COMPUTE

### B.1  LICENSES FOR MODELS AND DATASETS

As noted in section 3, we use two datasets for estimation - Tulu (Lambert et al., 2024) and C4 (Raffel et al., 2020) both of which fall under the Open Data Commons Attribution License (ODC-By) v1.0. Later we use MMLU Pro for behavioural evaluation (Wang et al., 2024) which falls under the Apache License (Version 2.0).

We study a wide array of models, below is license information grouped by model family:

- **OLMo:** The code and model are released under Apache 2.0.
- **Gemma:** Released under the gemma license stated here: https://ai.google.dev/gemma/terms
- **Llama:** Released under the llama license found here: https://www.llama.com/llama3/license/
- **Qwen:** The code and model are released under Apache 2.0.
- **Aya/Command:** Released under the Creative Commons Attribution Non Commercial 4.0
- **Pythia:** The code and model are released under Apache 2.0.

### B.2  COMPUTE RESOURCE AND COMPLEXITY

The estimation procedure used here has low complexity for an entropy estimator, requiring only a dot-product and softmax. The majority of compute expense comes from the model's forward pass required to compute the estimate. The complexity of this depends on the size of the model. In experiments here estimates required encoding 10,000 samples from C4 and Tulu. This process takes approximately 10, 40, or 70 minutes on either 2, 4, or 8 H100 GPUs respectively (number required depending on model size). Given this we estimate the total number of GPU hours required for the results in this paper at approximately 3,600 H100 hours.

## C  POST-TRAINING AND PREFERENCE INFORMATION

While LLMs become optimally compressed for next sequence prediction over pre-training, the final phase of the training pipeline often introduces other kinds of information. In the general case, post-training is designed to improve a model's ability to follow instructions and better align it with human preferences; we look at how this changes the information content of a model, and how it affects the representations from pre-training. Figure 7 shows preference information across two different families of open weights models, Llama and Gemma, which release a checkpoint at the end of pre-training and one at the end of post-training. In the Llama case post trained models consistently have higher preference information than their pre- and mid-trained counterparts. This supports a framing of pre-training as imbuing the model with core semantic information, which is later augmented with task-specific and preference information. With the Gemma models the picture is more complicated, with a consistent effect for the most recent Gemma 3 release - post trained models have greater preference information - but no significant pattern across earlier models.

## D  ADDITIONAL MODEL TIMECOURSES

A major challenge in studying pre-training is the limited availability of checkpoints. While there are a huge number of checkpoints available for final trained models, intermediate checkpoints over the course of pre-training are relatively rare. We focus analysis in the main paper on the OLMo2 models as they offer comprehensive check pointing – and comparatively strong performance. Here we look at two other families of models which make available some pre-training checkpoints. The Smol LM2 models (Allal et al., 2025) released this year are models with 1.7B parameters or smaller that achieve competitive performance. The 1.7B Smol model was trained on 11 Trillion tokens and performs comparably to the 1B OLMo2 model which was trained on 4 Trillion Tokens. Broadly the 1.7B Smol model follows a similar training trajectory to the OLMo2 1B model having phases of

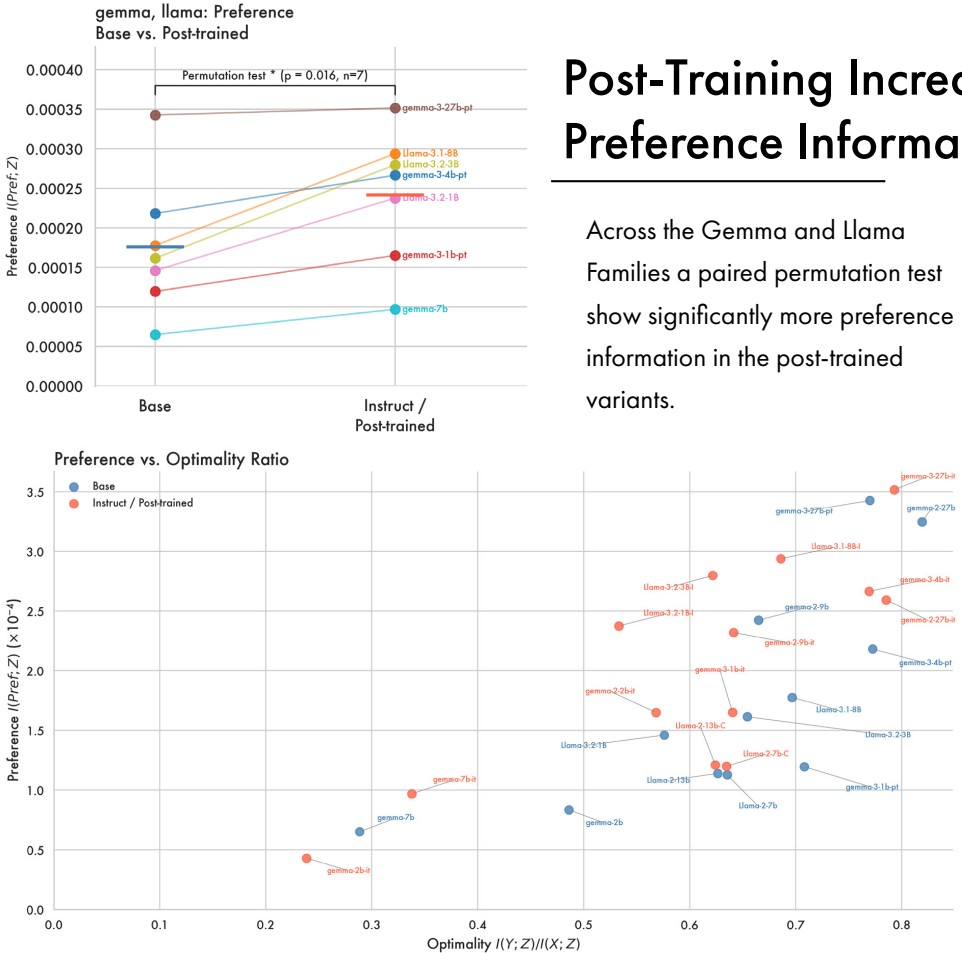

Figure 7: **Post-Training and Preference Information** (Left) Preference information on the vertical axis against whether or not the model is post-trained on the horizontal axis, with significance values from a paired permutation test above. (Right) Again, preference information on the vertical axis against optimality of a model's compression on the horizontal axis axis.

expansion and compression but failing to approach the bound like the OLMo2 7B and 32B models. Pretraining timecourses for the Smol 1.7B model are shown in Figure 8 with token, bigram, trigram, and quadgram backoff. This figure also includes trajectories for the smaller 100M and 400M variants - these mosels struggle to show much meaningful compression, though part of the issue may be that check-pointing starts comparatively late in the pre-trianing process compared with say the OLMo2 Models.

The other family of models we analyse are the Pythia models (Biderman et al., 2023). Timecourses for two Pythia models are shown in Figure 9. Included are analyses of the 1.4B and 6.9B models. In terms of parametrisation these are roughly comparable to the 1B and 7B OLMo2 models analysed in the main paper. However it's worth noting that the methodology for training these models is substantially different, and that their performance is substantially lower than the OLMo2 models, and other more recent open-weights models analysed above. In terms of training, Pythia models are intended for scientific analysis, as a result they use the same amount of data, batch size, and number of training steps across model sizes. Perhaps most importantly these models are trained on the Pile dataset (L. Gao et al., 2020). This contains roughly 299,892,736,000, by contrast the 1B OLMo2 model is trained on 4,000,000,000,000 tokens - meaning the Pythia models see 7.5% of that data. Accordingly the 1.4B Pythia model appears to achieve better compression later in training than its OLMo counterpart. As discussed in the main paper there may be a relationship between data

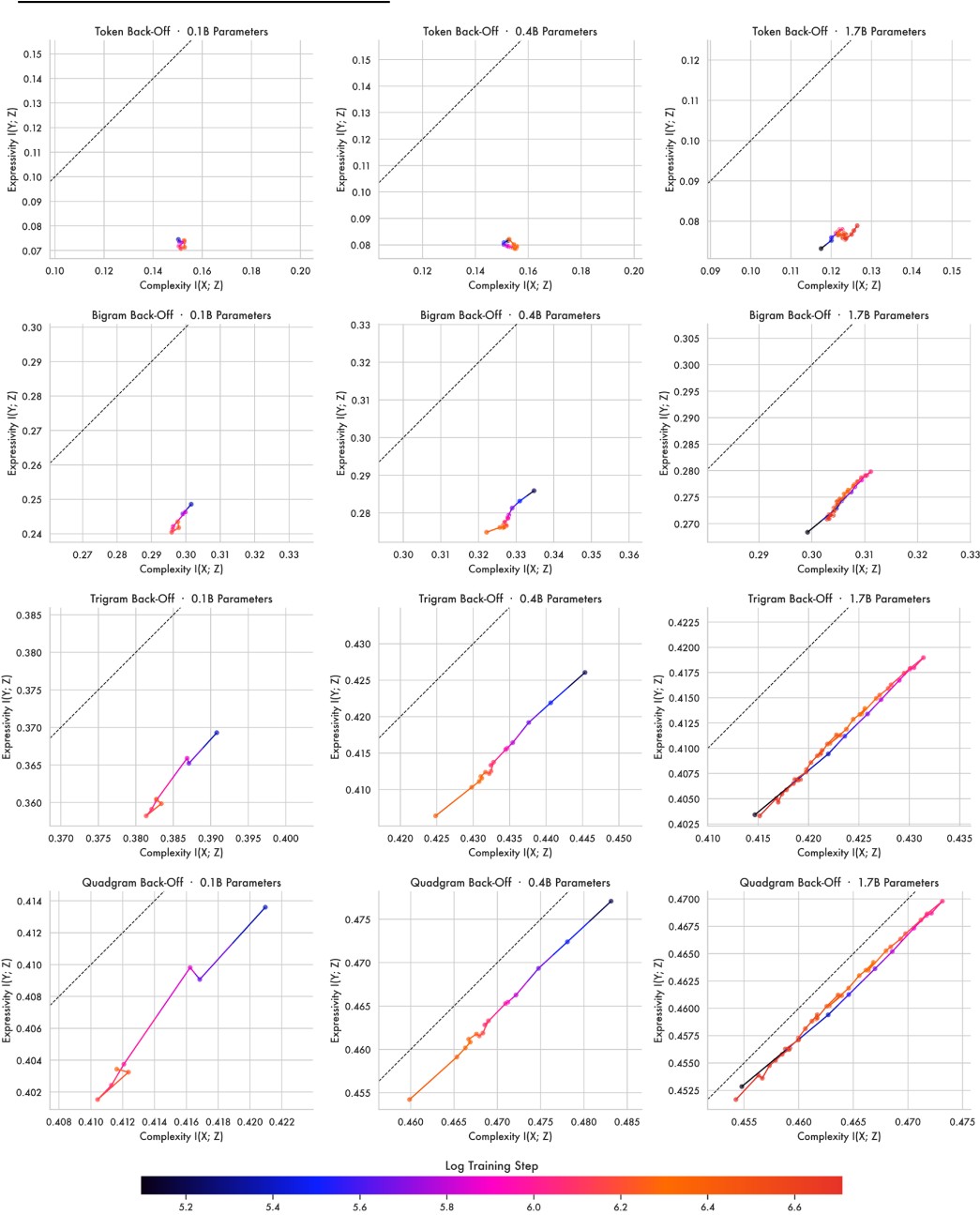

Figure 8: **Smol LM2 Timecourses**

complexity and the model complexity needed in order to achieve substantive compression of it. By contrast the 6.9B Pythia model is still expanding representations late into pre-training; this would appear to indicate it is under-trained.

# Pythia Pre-Training Timecourse

### 1.4B, 6.9B parameters

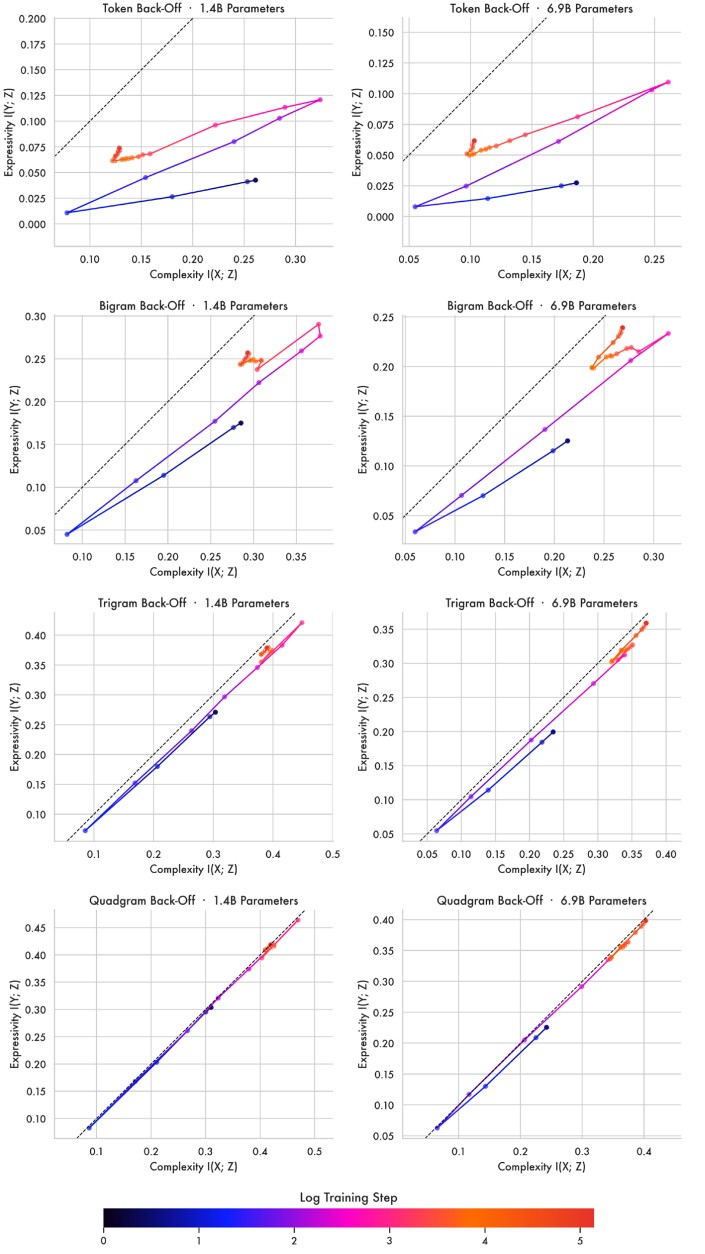

Pythia models are trained on orders of magnitude fewer tokens than the OLMo2 Models shown in the main paper - only 7% of the OLMo2 data. Additionally unlike other LLM families all sizes see the same amount of data, in the same batch size, for the same number of steps.

Accordingly here, the Pythia 6.9B model appears undertrained, still expanding representations even late in training. Having seen the same amount of data the smaller model follows the expected two phase trajectory. Here the 1.4B pythia model may achieve better compression given it is trained on only a small fraction of the data OLMo2 1B is expected to compress.

Figure 9: **Pythia Model Timecourses**

## E    ENTROPY ESTIMATION

### E.1    ESTIMATOR HYPERPARAMETERS

The estimator has two parameters that need to be set the number of bins $m$ and the temperature used in the softmax $\varepsilon$. As shown appendix G.2 and (Conklin, 2025), the estimator is generally robust with respect to the number of bins - in all experiments presented here we use $m = 100$.

### E.1.1 TEMPERATURE CALIBRATION.

Naively we could use the same temperature across all models, however models broadly differ in the dimensionality of their hidden representations. Within self-attention as the dimensionality $d_k$ of query and key vectors grows, the variance of their dot products scales linearly with $d_k$, pushing the softmax function into saturated regions where gradients vanish (Vaswani et al., 2017). This is a specific instance of the broader concentration of measure phenomenon in high-dimensional spaces, where distance and similarity metrics become increasingly uniform and less discriminative (Aggarwal et al., 2001). As a result Vaswani et al. (2017) scales the dot product by $\sqrt{d}$ to avoid saturation. The soft entropy estimator uses dot-products on the surface of a the unit hypersphere passed through a softmax in order to estimate a density. As a result, for a fixed temperature, higher-dimensional space will begin to appear more uniformly distributed. We compute a temperature which is calibrated to prevent saturation making estimates of for different dimensionalities directly comparable.

Let $V_\varepsilon$ denote the von Mises–Fisher (vMF) distribution on $\mathbb{S}^{d-1}$ with concentration parameter $1/\varepsilon$; by rotational symmetry, $D_{\mathrm{KL}}(V_\varepsilon \| U)$ depends only on $\varepsilon$ and $d$, and measures how far a single vMF kernel is from uniform. For an arbitrary data distribution $P$ on the sphere, let $P_\varepsilon$ denote the convolution of $P$ with the vMF kernel at temperature $\varepsilon$; the smoothed KL divergence $D_{\mathrm{KL}}(P_\varepsilon \| U)$ is the estimation target, and satisfies $0 \le D_{\mathrm{KL}}(P_\varepsilon \| U) \le D_{\mathrm{KL}}(V_\varepsilon \| U)$ .

For the soft entropy estimator $\widehat{D}^{(\mathrm{SQ})} = D_{\mathrm{KL}}(\hat{p} \| u_m)$ takes values in $[0, \log m]$, where $m$ is the number of bins. To ensure this range is well-matched to the estimation target $D_{\mathrm{KL}}(P_\varepsilon \| U)$, we calibrate the temperature $\varepsilon$ so that the maximum possible value of the target equals $\log m$. The target is bounded above by $D_{\mathrm{KL}}(V_\varepsilon \| U)$, the KL divergence of a single von Mises–Fisher kernel from uniform on the unit hypersphere $\mathbb{S}^{d-1}$. Direct computation of $D_{\mathrm{KL}}(V_\varepsilon \| U)$ requires evaluating modified Bessel functions, which is numerically unstable at large $d$; however, using Amos-type bounds on Bessel function ratios (Amos, 1974), one can construct upper and lower envelope functions $\Psi^{\pm}_{\varepsilon,d}$ satisfying $\Psi^{-}_{\varepsilon,d} \le D_{\mathrm{KL}}(V_\varepsilon \| U) \le \Psi^{+}_{\varepsilon,d}$, with a gap of order $O(d^{-1})$. Both bounds are strictly monotone decreasing in $\varepsilon$ for $d > 4$, so the equation $D_{\mathrm{KL}}(V_\varepsilon \| U) = \log m$ has a unique solution. To leading order in $d$, this solution is $\varepsilon^\star(m, d) = 1/\sqrt{2d \log m}$, which we use throughout our experiments.

$$\varepsilon^\star(m, d) = \frac{1}{\sqrt{2d \log m}}. \tag{7}$$

In practice this is computed once per model based on $m = 100$ and the model's dimensionality. The correction here bears strong resemblance to the default scaling within self attention $\sqrt{d}$.

### E.2 APPROXIMATING THE INPUT DISTRIBUTION

We estimate the mutual information between model inputs and outputs. In an auto-regressive decoder-only LLM the input to a model is the preceding context up to the current token. We view the input as n-grams of tokens where the input at timestep $x_t$ is an ngram of width $t$ containing all tokens $x_0...x_t$. Maintaining probability distributions for every possible context proves intractable due to the combinatorial complexity of natural language. Additionally, ngrams greater than 3 tokens become sparsely distributed in the data making reliable estimation of their probabilities a challenge. As a result we condition estimates on ngrams of fixed-widths 1, 2, 3, 4 - referred to in the paper as token, bigram, trigram, quadgram. This is related to Backoff (Katz, 1987) which reduces n-gram size until the n-gram has non-zero probability in a corpus. Here though we do not interpolate different n-gram widths, instead maintaining separate aggregate estimates for each width – in part to be able to study how different levels of contextual information are represented in the model. Where a given n-gram, like a quadgram, does not have non-zero probability in the data it is omitted from overall quadgram mutual information estimate.

In practice this means estimates for smaller n-gram widths are more reliable - a classical issue in language modelling (see Jurafsky & Martin, 2000, p.32). Token, bigram, and trigram estimates can be estimated reliably from a relatively small sample of data. We judge this by looking at how estimates change as a function of the number of samples during the estimation procedure, by 5,000 samples these estimates relaibly begin to converge. Quadgrams, due to their sparsity, tend to have less robust

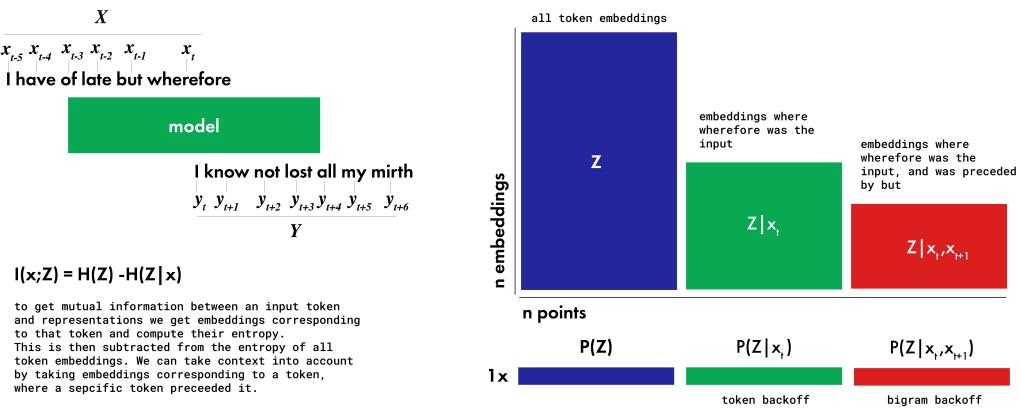

Figure 10: **Illustration of conditional probability estimates.** An example sentence is provided, assuming word-level tokenization for simplicity. At left are the indices for the input and output tokens when the current input word is *wherefore*. At right is shown the sub-setting procedure for estimating conditional probabilities. This illustrates how bigram estimates do not compute entropy of two token embeddings, rather the embedding for the current token embedding conditioned on preceding context.

estimates - additionally the number of labels grows quadratically with each additional ngram width making quadgrams challenging to estimate for larger models with larger vocabularies. As a result our broad comparison of open-weights models uses token and bigram estimates. Additionally the pre-training model size analysis here focusses on trigram estimates (figure 3) as this widest context that still reflects a reliable estimate. The analysis of how context is represented over pretraining (figure 2) includes quadgram estimates for reference.

### E.3    APPROXIMATING THE OUTPUT DISTRIBUTION

While during inference models predict the next token given preceding context, this is distinct from how they are trained. During training of an auto-regressive decoder-only LLM causal masking means a token can only attend to preceding context, not trailing context. However transformer decoders are trained using teacher forcing, where predictions are generated for the entire sequence in parallel by assuming predictions are made correctly. This is instead of having training operate on one token at a time with a separate forward pass for each - which is how predictions are generated during inference. The result of this is that for an embedding $e_t$ at timestep $t$, following embeddings $e_{t+1}$ can attend to $e_t$. This means embeddings get gradient information from the trailing context. Put another way, the prediction for output $y_{t+1}$ is written in terms of $e_t$. As a result the gradient information from the next token(s) in a sequence $\nabla\mathcal{L}_\theta(y_{t+1})$ affect the embedding at the current timestep.

Given that our analysis computes embedding mutual informations over training with respect to a model's input and outputs this fact has implications for us. It means that the output for $e_t$ is not just $y_{t+1}$ but all following output tokens $y_{t+1}...y_n$ where $n$ is the sequence length. This is because $e_t$ receives gradient information from the loss with respect to predicting all following tokens in a sequence. As a result we consider $X$ to be the entire preceding context in the input (as mentioned above), and $Y$ to be the entire trailing context after the current point in the sequence. This means when we compute mutual informations for different n-gram widths we match the width for $X$ and $Y$ - conditioning the estimates on the same width of preceding and trailing context respectively.

### E.4    ESTIMATING MUTUAL INFORMATIONS

To compute mutual informations between the input $X$ and representations $Z$, we need two quantities: the entropy of representations $\mathcal{H}(Z)$ and the conditional entropy given the input $\mathcal{H}(Z|X)$. To

compute $\mathcal{H}(Z)$ we use the quantisation procure described in section 3.1 applied to all token embeddings which gives $\hat{Z}$ - by summing over the each embedding and renormalising we get a categorical distribution $P(Z)$ that describes the embedding space. To get a conditional estimate $P(Z|X)$ we simply take $\hat{Z}$ and compute a subset, containing the embeddings corresponding to the input $X$, $\hat{Z}|X$. Summing and renormalising gives us the distribution $P(\hat{Z}|X)$.

This brings us to an important distinction, our analysis discusses mutual informations with respect to tokens, bigrams, trigrams, and quadgrams. These are not computed over different widths of embeddings, but rather over single token embeddings conditioned on the preceding context. In the same way $P(\hat{Z}|X)$ is computed as a subset of $P(Z)$, as we condition on further context we can subset the embeddings further. Figure 10 gives a high-level illustration of this process. It means that $Z|$token is a subset of $Z$, and $Z|$bigram is a subset of $Z|$token, $Z|$trigram is a subset of $Z|$bigram etc. This means the terms token, or bigram mutual informations refer to the width of the conditioning context, not the width of the embeddings over which entropy is computed.

### E.5 Conditional Mutual Informations and The Residual

In order to compute what proportion of a model encodes each level of context we use the chain rule for mutual information. As we increase the context width used in back-off the estimates contain each other - the bigram mutual information includes the token mutual information. This means when estimating mutual information with the source distribution with token back-off we look at only the current token $I(Z; x_t)$, while bigram mutual information adds the preceding token $I(Z; x_t, x_{t-1})$. The chain rule here means:

$$I(X; x_t, x_{t-1}) = I(Z; x_t) + I(Z; x_{t-1}|x_t). \tag{8}$$

which allows us to separate out the information explained by the current token $I(Z; x_t)$ and the preceding one given the current token $I(Z; x_{t-1}|x_t)$. For the source distribution $X$ and a given n-gram width $n$ we can get a proportion $\phi$ of model information by normalising by the entropy of the model.

$$\phi(x, n) = \frac{I(Z; x_n|x_1..x_{n-1})}{\mathcal{H}(Z)} \tag{9}$$

We compute this for each level of backoff, where at the token level $\phi(x, 1) = I(Z; x_t)/\mathcal{H}(Z)$. The most granular label we have in the experiments here is quadgram. The residual, or unexplained information, is the information in the model left after subtracting the mutual information of the most granular category.

$$\phi_{residual}(x, n_{max}) = \mathcal{H}(Z) - I(Z; x_n..x_{n_{max}}) \tag{10}$$

Results showing proportions of model information broken down in ngram can be found in figure 2 (bottom).

#### E.5.1 Conditional Mutual Informations and Performance

Shown in Figure 11, less token information, but a higher proportion of local contextual information relates significantly to downstream performance. In the main paper we report token level back-off in the correlation results, where lower complexity is related to performance.

### E.6 On The Use of Shannon Entropy

In this paper we compute the entropy of continuous latent variables. As a result it is natural to ask why we - in line with previous work (Sajjadi et al., 2018; Shwartz-Ziv & Tishby, 2017; Voita et al., 2019) - opt instead to discretise representations and compute their Shannon entropy (Shannon, 1948). There are two major reasons for this; first, differential entropy is not the true continuous analogue of Shannon Entropy (Jaynes, 1957). This is shown by the fact that differential entropy

# Context vs. Performance

## More context information relates to performance

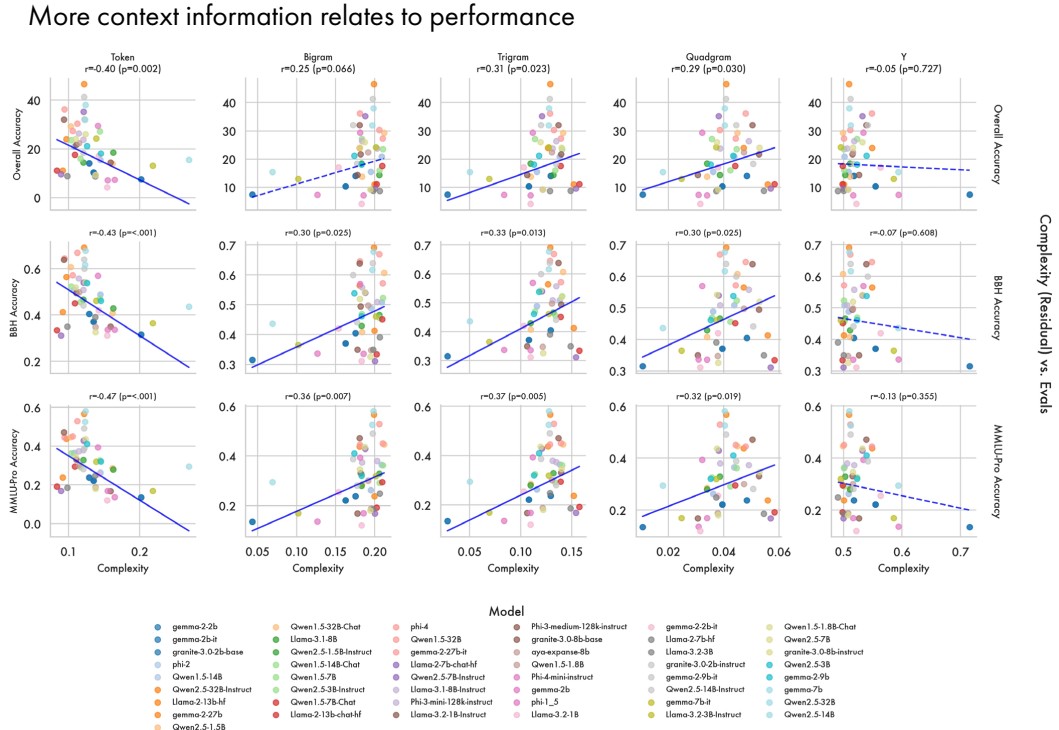

Figure 11: **Proportions of Information Vs. Performance:** Across 47 models less token information and more local contextual information relates signifcantly to performance based on a spearman correlation (significance indicated by hue).

$D(X)$ is unbounded $-\infty \le D(X) \le \infty$, and variant under linear transformations. This is the main motivator for an information theoretic analysis to discretise and use Shannon entropy directly. A secondary consideration is that we don't know how embeddings are distributed, so in order to get a differential entropy estimate we would first need to fit a distribution to the data. At scale this fitting step can be expensive, and introduce topographic assumptions. While discretisation is imperfect it enables the use of Shannon entropy, and makes minimal topographic assumptions.

### E.7 SCALABILITY OF PRIOR WORK

Notably Shwartz-Ziv and Tishby (2017) perform an empirical information theoretic analysis of neural-networks trained on MNIST. To do so they perform dimension-wise discretisation of model embeddings. This turns a 16-dimensional vector into a 16 character string. They then convert this to a categorical distribution over all possible strings. This gives a single categorical distribution that can describe representations at a particular layer in a network. This approach to discretisation works well on small problems - they study feed-forward networks trained on MNIST. However the dimension-wise discretisation requires taking a hidden representation with dimensions *batch* × *hidden* and transforming it to *batch* × *hidden* × *n bins*. If using 50 bins, in practice this means using 50 times the memory of not discretising. For the OLMo2 32B model used in this paper which has a hidden dimension of 5120 and 64 layers, and where we have a context window of 512 tokens, this would require holding in memory a tensor of dimensions *batch* × 512 × 5120 × 44 × *n bins*. The memory use of this approach makes it intractable to apply to contemporary models and the problems studied here.

Voita et al. (2019) studied the transformer base model which has only 6 layers with a hidden dimension of 512. Despite this they note the approach from Shwartz-Ziv and Tishby (2017) was not

tractable to apply to the model. They opt instead for quantising representations via clustering, based on related work from (Sajjadi et al., 2018). This method runs a clustering algorithm (Voita et al. (2019) use mini-batch k-means), then treats each cluster as an event in a categorical distribution - where density is assigned proportional to cluster membership. While this method provides robust entropy estimates, and dramatically less memory usage than the approach from Shwartz-Ziv and Tishby (2017) it still has relatively high computational complexity. It requires running a clustering algorithm to convergence, before performing quantisation prohibiting its use in an online setting - you need the cluster centroids before you can assign embeddings to them. Again thinking of the OLMo2 32B model used here, this would require running a clustering algorithm on 5120 dridemimensional spaces, at all 44 layers separately, for each of the 150 pre-training checkpoints. This would provide the 'bins' for the quantisation, then embeddings would need to be assigned to bins, requiring a second forward pass.

In practice an information-theoretic analysis of an LLM requires an entropy estimation method that is memory efficient, fast to compute, and can be applied in an online setting - requiring a single forward pass and no caching of the embeddings. The only estimator we're aware of that meets these criteria is the soft-entropy estimator (Conklin, 2025). Here the quantisation requires only a cosine-similarity and a softmax making it fast and memory efficient. Additionally the normalisation step means 'bins' can be computed once at the start of the analysis, rather than needing a pass through the data to fit clusters or fit the support of the model's distribution. Conklin (2025) notes that the use of cosine similarities means this method considers only angular information in the representation space. While euclidean distances can be used instead, this would require first estimating the support of the distribution to fit the 'bins' making online estimation challenging. However use of cosine-based methods is standard practice in NLP (T. Gao et al., 2021; Reimers & Gurevych, 2019; Zhang et al., 2020), with some work suggesting vector norms in LLMs predominantly encode frequency information (e.g. Oyama et al., 2023).

### E.8 THE INFORMATION BOTTLENECK BOUND

The Information Bottleneck bound is the curve traced by varying the trade-off parameter $\beta$ in:

$$\mathcal{F}_\beta[p(Z|X)] = I(X;Z) - \beta I(Y;Z) \tag{11}$$

The curve this traces is where representations are optimally compressed. Along this bound $p(Z|X)$ is an optimal encoder, from inputs to representations, preserving only the information in $X$ relevant to $Y$. For a given dataset this optimal encoder can be found numerically via a version of the Blaut-Arimoto (Arimoto, 1972; Blahut, 1972) method for computing channel capacity. Introduced in Tishby et al. (2000), the information bottleneck method for determining channel capacity relies on three equations:

$$p_\beta(z|x) = \frac{p_\beta(z)}{Z_\beta(x)} \exp\left(-\beta D[p(y|x)||p_\beta(y|Z)]\right) \tag{12}$$

$$p_\beta(z) = \sum_{x \in X} p(x)p_\beta(z|x) \tag{13}$$

$$p_\beta(y|z) = \sum_{x \in X} p_\beta(x|z)p(y|x) \tag{14}$$

These equations are satisfied self-consistently at the bound. As these three equations rely on each other one can learn an optimal encoder by starting with a randomly initialised one. Then iteratively computing each of these equations in turn.

In the general case the shape of this bound follows a linear relationship, until all mutual information between $x$ and $y$ is captured. At this point the curve saturates — additional complexity doesn't result in additional accuracy, as there's no more predictive information in $x$. This means numerical computational bound is largely important for computing where it saturates.

*Pre-Training Loss vs.*
*Compression Optimality*

During pre-training models experience dramatic reductions in loss early on before this begins to saturate.

As the loss begins to saturate the representations begin to approach the bound.

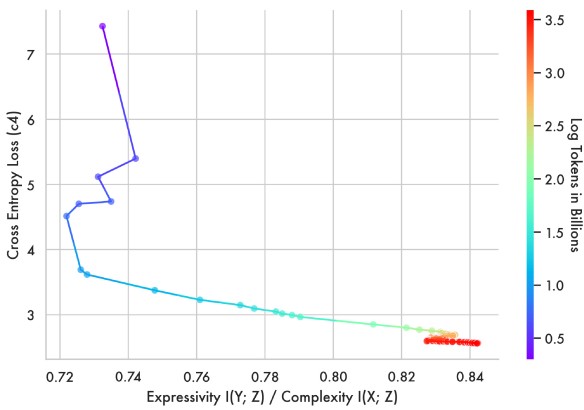

Figure 12: **Cross-Entropy Loss vs. Distance to bound.** Shown on the vertical axis is the OLMo2 7b model's cross-entropy loss on 10,000 examples from c4. On the horizontal axis is the ratio between $I(Y; Z)$ and $I(X; Z)$ which is indicative of how close a representation is to the IB bound. Models begin to compress and approach the bound as the loss saturates.

However numerical computation of the bound in our setting proves intractable. Here the optimal encoder $p(z|x)$ needs to map all of natural language to representations that optimally predict the next token. This is an exceedingly challenging problem for an iterative numerical optimizer – it's a problem that ordinarily requires a large language model. In experiments we are able to compute a bound for tokenizers up to 50,000 tokens, however past this point convergence begins to fail. In our setting this process takes a tokenizer and 300,000 sentences from c4 to get a maximum likelihood estimate of $P(x), P(y), P(y|x)$ on data representative of a model's training data. We can then iteratively compute $P(z|x)$ until convergence. In our experiments this iterative procedure converges to the expected saturation point - where $I(Z; Y) = I(X; Y)$.

Given that we would like to have a bound for problems where numerical computation of it proves intractable, we leverage this pattern by assuming the bound follows a linear relationship until the saturation point where $I(Z; Y) = I(X; Y)$. The largest tokenizer for which we can tractably compute this quantity has a normalised $I(X; Y)$ of 0.7 (where 1.0 is the maximum possible value). Across all open weights models the highest token complexity converged to is 0.15, well below the saturation point. This is in line with results from (Shwartz-Ziv & Tishby, 2017), which shows FFNs on MNIST only converge near the saturation point when over-fitting.

# F    RELATING COMPRESSION TO TRAINING LOSS

Prior work (Shwartz-Ziv & Tishby, 2017) shows that models transition from the fitting phase where $I(Y; Z)$ increases, to the compression phase where $I(X; Z)$ decreases when empirical error on the training distribution saturates. Their setting is substantively different to the one studied in our work – the most relevant differences here are that they analyse a feed-forward model trained on MNIST for multiple epochs, meaning the model's performance can fully saturate in-distribution. In an LLM setting models are trained on orders of magnitude more data, often for a single epoch - passing through the data only once, meaning saturation is more graded. However it is worth asking if the transition between fitting and compression relates to training performance in an analogous way.

We compute the cross-entropy loss for the OLMo2 7b model performing next token prediction on 10,000 examples from C4. C4 is a substantive component of the OLMo2 pre-training data (OLMo et al., 2025) and so gives us a proxy for in-distribution performance on the model's training set. This follows a previously attested dynamic, where earlier steps dramatically decrease the loss before this begins to slowly saturate. Unlike in an MNIST setting this objective never truly saturates, instead slowly flattening. Figure 12 shows this loss plotted against the ratio between expressivity $I(Y; Z)$ and complexity $I(X; Z)$. As noted in section 4.2 this ratio acts a distance to the bound where models are optimally compressed - as this quantity approaches 1.0 models approach the bound. Figure 12

*Robustness to Number of Reference Points $W_i$*

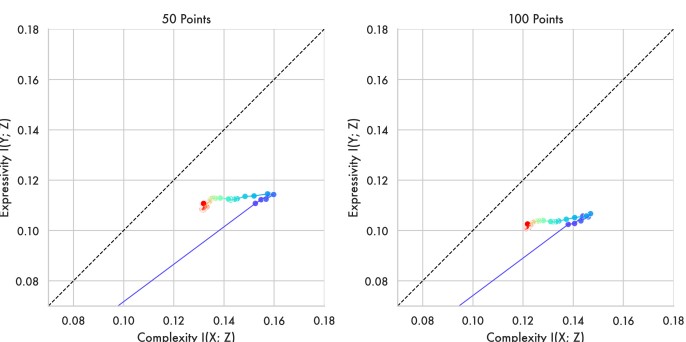

*Robustness to Data Distribution*

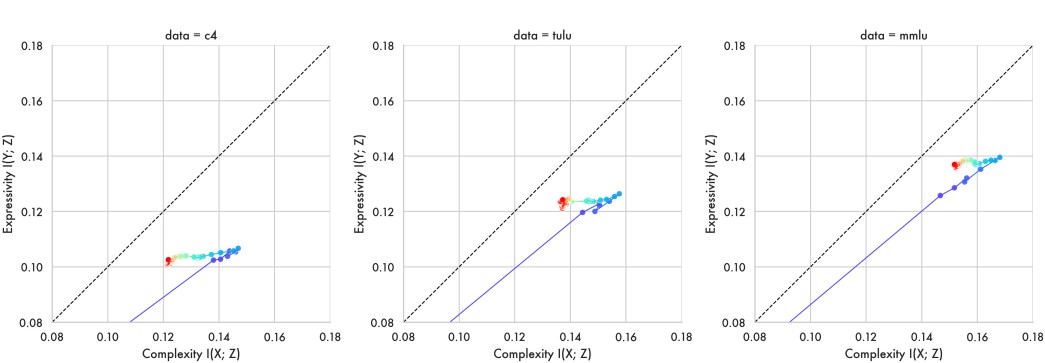

Figure 13: **Estimator Robustness to number of bins and data distribution.** Shown are trajectories through the information plane for the OLMo2 7b model. (Top) trajectories in the main paper use 100 reference points $w_i$ per layer, here 50 points are used, and show the same overall two-phase pattern. (bottom) estimates in the main paper are with respect to c4 given its resemblance to an LLM's training distribution, here are show trajectories computed with data from tulu and MMLU pro which show the same broad two-phase pattern. Hue indicates log tokens in billions over the course of pre-training.

shows how models begin to approach the bound as the loss on c4 begins to saturate. This broadly aligns with Shwartz-Ziv and Tishby (2017), where saturation in-distribution relates to the transition to compression.

## G ESTIMATOR ROBUSTNESS

Our work does not introduce the soft entropy estimator but is the first to apply it in this context. As a result we run some robustness experiments to see how the results vary under different hyper-parameters and data distributions.

### G.1 ROBUSTNESS TO DATA DISTRIBUTION

Core results in the paper show information plane trajectories computed using C4 as this dataset forms a substantive part of the pre-training data for the OLMo2 models. To verify that the overall pattern of expansion and compression is robust across data distributions we analyse the pre-training checkpoints of the OLMo2 7B model across data from C4, Tulu (Lambert et al., 2024), and MMLU (Hendrycks et al., 2020). Figure 13 (bottom) shows the pre-training trajectories for the 7B model

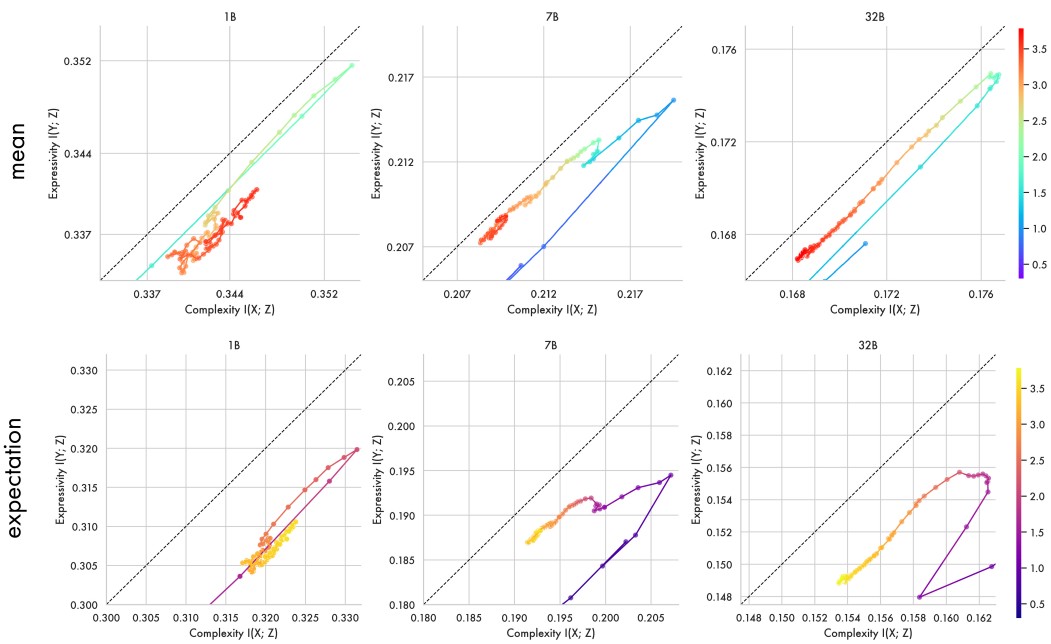

Figure 14: **Pre-training Time-courses Computed with a Mean vs. Expectation** Shown are trajectories through the information plane over pre-training for the OLMo2 1b, 7b, 32b models. These analyses use a mean (top) or expectation (bottom) in computation of mutual information. The expectation is used in the main paper as it reflects true mutual information. Hue indicates tokens in billions over the course of pre-training.

computed with token backoff across all three distributions. The broad two-phase pattern proves consistent across all of them with a fitting phase followed by a compression phase where models approach the bound. There are individual variations for each dataset, with Tulu and MMLU having higher mutual informations than C4. This may reflect that MMLU and Tulu are more domain-specific than C4 which is a broad crawl of the internet.

### G.2 ROBUSTNESS TO NUMBER OF REFERENCE POINTS

The Soft Entropy estimator relies on a soft-quantisation of a model's embedding space, whereby each representation is softly assigned to $n$ points $w_i$ which are sampled uniformly at random from the surface of the unit sphere (a process described in equations 2 and 3). Experiments in the paper use a $n = 100$. Here we show the core 7b model pre-training time-course computed for c4 with token backoff using $n = 100$ and $n = 50$. The results show the same overall pattern of expansion and compression with small changes to the exact mutual information values. Given this estimator resembles a differentiable relaxation of a binning-based estimate, it is relevant to note that in binning based approaches increasing the number of bins reduces mutual information by assigning similar representations to an increasing number of different bins (Paninski, 2003). The results seen here are consistent with this effect with 100 points achieving slightly lower mutual informations than 50 points Conklin (2025) note when introducing this estimator that this interaction can occur, but in benchmarking experiments show the soft-assignment process makes this estimator more robust to number of 'bins' than existing clustering based approaches.

### G.3 LANGUAGE IS A LONG-TAILED DISTRIBUTION — COMPUTING 'MUTUAL INFORMATION' WITH MEANS

As noted in section 3.1, the quantity estimated here is mutual information which uses an expectation over conditional entropies rather than a mean.

$$I(X;Z) := \mathcal{H}(Z) - \sum_{x \in X} P(X = x)\mathcal{H}(Z|X = x) \tag{15}$$

Here we recompute the core pre-training analyses for the 1B, 7B and 32B models using a mean - detailed in equation G.3 - to see how treating each event as equiprobable effects the analysis. Given language is known to be zipfian distributed a small number of high-probability patterns likely drive the mutual information. It is worth noting when using a mean the resulting quantity is not the true mutual information, and so the information bottleneck bound does not necessarily apply.

$$I(X;Z) := \frac{1}{|X|} \sum_{x \in X} \mathcal{H}(Z) - \mathcal{H}(Z|X = x) \tag{16}$$

As shown in figure 14, estimates computed with the mean, and with the expectation show the same broad two-phase pattern. with models first expanding representation phase before compressing towards the bound. When taken as a mean the quantity reflects the mean mutual information per label - like mean mutual information per token - rather than being weighted by the exponentially distributed token representations.

