# OpenReview forum: "Learning is Forgetting; LLM Training As Lossy Compression"
_ICLR.cc/2026/Conference — ICLR 2026 Poster_

### Official Review · Reviewer_FQUJ · 2025-10-22

**Soundness:** 3
**Presentation:** 4
**Contribution:** 3
**Rating:** 6
**Confidence:** 4

**Summary:**

This work applies the idea of information bottleneck to study LLM training. The authors propose an entropy estimation method and use it to quantify the mutual information between the representations and inputs/targets. By studying these quantities, they conclude that LLM training consists of two phases: a fitting phase and a compression phase, so training can be viewed as lossy compression. They further show that their framework reveals a correlation between model performance and the optimality of the compression.

**Strengths:**

1. The work applies the interesting idea of information bottleneck to study models/datasets at much larger scale than prior works.
2. A tractable method for estimating entropy at LLM scale is used.

**Weaknesses:**

1. The existence of two phases is not clear from the figures. How do we know that the change in I(Y; Z) and I(X; Z) is significant?
2. In Figure 1, the two phases on the left and right don’t quite look similar.
3. If the changes are indeed significant, then the compression phase seems to consist of a significant decrease in expressivity (mutual information with the label). This casts doubts on a) whether the entropy is estimated accurately enough, and b) whether the “compression phase” is really about compressing irrelevant information between X and Z.
4. In Figure 1, the x and y axes have numbers between 0.2-0.22. Doesn’t that imply that there’s barely any difference in the mutual information throughout training? In other words, how do we know that the differences are significant?
5. Commonly used language datasets are highly imbalanced. While I understand the authors’ desire to avoid weighing tokens like “the” and “a” too heavily, I question whether weighing everything equally skews the estimate too much.

**Questions:**

1. In the context of the information bottleneck, there were several papers that questioned the relation between compression and generalization. Could the authors add a literature survey on that topic and elucidate what those papers got wrong?
2. Is the method used for estimating mutual information similar to the Nystrom method in kernel methods?
3. Is there any prior work that also measures compression of token embeddings and looks into whether such properties correlate with generalization?
4. Do we have a reason to expect that the method would actually work once the number of points w_i is small (which is required for estimating mutual information based on the resulting softmax probabilities)?
5. Did you conduct experiments testing the robustness of observations to the number of points w_i in eq. 2?
6. Does expressivity alone predict performance? What is the correlation between them?
7. How do you expect the trajectories to change as the context length used to estimate entropy increases?
8. Does data quality impact the presence of the two phases?
9. In the original IB paper, the "knee" in the curve was due to the interpolation of the training data (transition from getting zero error on ERM to compression). Could you show that the same thing occurs in your experiments?

---

> ### Author Response · Authors · 2025-11-26
>
> Thank you for your engagement with our work, and for taking the time to raise issues that will make the paper stronger. We first summarise the changes made to the submission in response to your feedback before responding to questions and weaknesses point by point.
>
> - In response to asking if expressivity alone predicts performance we have added correlation results relating expressivity and complexity independently to MMLU Pro performance to the main paper (figure 4). These show that only their ratio - our distance to bound measure - has a significant relationship with performance.
>
> - We have added an analysis relating a model’s loss on next token prediction to our distance to bound measure in figure 1 of the main paper and Appendix F. We show the OLMo2 7b model enters the compression regime approaching the bound on optimal compression as the training loss saturates. This is an analogue to the previous work you mention relating “zero error” to compression. (this required computing the entropy analysis for 106 additional 7B model checkpoints)
>
> - We have added a robustness analysis (Appendix G) which shows the core two-phase pattern of training is robust to the number of points used in W as well as the data distribution. This required re-running the pretraining timecourse of the 7B model with half the number of points, and with data from C4, Tulu, and MMLU (this required computing the entropy analysis for 424 additional 7B model checkpoints)
>
> - In response to concerns about our taking mean label-level mutual informations rather than an expectation, we have added the pretraining analysis for the OLMo2 1B, 7B, and 32B models computed with an expectation to appendix G.3 showing this does not substantially change the pretraining trajectory through the information plane. (This required computing the analysis for 283 different model checkpoints of various sizes)
>
> ## Questions
>
> 1. *Have you discussed prior work on compression in the IB context?*
> We do mention the work that discusses the relationship between compression and generalisation in the IB in the related work section (the paragraph starting at line 169). This discussion largely centers around the validity of the compression analysis in the work from Shwartz-Ziv and Tishby (2017). In their analysis they consider a feed-forward network trained on MNIST with tanh non-linearities. Saxe et al. (2018) observe that models with these non-linearities naturally saturate over training and that this may be an artifact of the non-linearity rather than a general property of deep-learning. Goldfied et al. (2019) is the primary citation for claiming compression in an IB context does not necessarily relate to generalisation. We agree that it is important to reference these findings, but given that this previous work exclusively studies MNIST and Feed Forward networks - which has limited relevance to the objects of study here - we keep this discussion to one paragraph in the literature review.
>
>
> 2. *Is your estimator related to the Nystrom method?*
> This is a nice comparison to make, but the two methods are distinct. The Nystrom method approximates an N x N kernel by subsampling q examples and computing an N x q matrix of pair-wise distances. The soft entropy estimator computes an N x M matrix of distances, where M are points sampled uniformly at random from the surface of the unit sphere. Critically here M is not a subset of samples in N, rather a set of reference points used analogously to bins. So both methods involve computing distances between two sets of points, but in the Nystrom method N is compared with a subset of N, while in the soft-entropy estimator it is compared with a set of points sampled from a reference measure.
>
> 3. *Has embedding compression been related to performance?*
> We are not aware of work that relates rate-distortion compression to performance in LLMs. As noted in the related work (line 120) the bias-variance tradeoff (Geman et al., 1992) relates model complexity to generalisation, but this is a theoretical result from well before LLMs. In the literature review we do note recent mechanistic interpretability work (line 199) that has used cross-coders to characterise when features emerge, and who argue for a similar two-phase training dynamic however without any quantification of compression.
>
> 4. *Will the estimator work with a small number of points?*
> The work that introduces the soft entropy estimator (Conklin, 2025) presents benchmarking showing how the number of points w_i affects the accuracy of the estimate. In their work a very small number of points have an adverse effect on the quality of the estimate, but past 25 points-per-layer estimates are robust, even for high dimensional spaces.

---

> > ### Author Response · Authors · 2025-11-26
> >
> > ## Questions (cont.)
> >
> > 5. *Have you tested robustness of the estimator to number of points?*
> > We have added to the appendix a replication of the OLMo 2 7b pre-training analysis using 0.5x the bins of the original paper to show the overall trajectory is robust regardless of the precise number of points - this can be found as part of the robustness analyses in appendix G.2.
> >
> > 6. *Does expressivity alone predict performance?*
> > This is an excellent point that we would love to clarify. We have analysed whether or not expressivity I(y;z) alone is a significant predictor of downstream performance, and it is not. Neither complexity, nor expressivity alone has a significant relationship with performance in our experiments - it is instead their ratio which proves significant. This is the basis for our arguing that distance to the IB bound (what we term optimality) is what matters, rather than either quantity in isolation. In the original submission we included only the results regarding complexity, optimality, and preference information. We have updated these results to include the null result for relating expressivity and downstream performance in figure 4.
> >
> > 7. *How does context length affect compression?*
> > The relationship between context length and compression is an interesting question for future work. We expect different context lengths to have minimal effect on the core estimates presented here. This is because of the results discussed at line 369 and shown in figure 2. These show that as the mutual information estimate is conditioned on wider context (token -> bigram -> trigram) the mutual information values begin to converge. Most of the information in an embedding is explained in terms of local contextual information, likely reflecting the short dependency lengths of the natural language on which these models are trained. As a result increasing the context window will likely have minimal effect on the core results here. In future work we hope to systematically study how long-context models differ in terms of the compressions they converge to.
> >
> > 8. *Does data quality affect presence of two-phases?*
> > We have run analyses with different splits of data, for example we computed basic complexity and expressivity scores for pretraining with respect to Tulu preference data and MMLU - these can be found in appendix G.1. Results show that the exact amount of mutual information varies based on the data used to compute the estimate, however the overall pattern of expansion and compression does not. This suggests to us that the core phenomena observed are robust across different data distributions.
> >
> > 9. *The knee in the original IB paper was related to error saturating, is that the case here?*
> >  Previous work on the IB applied to deep learning - Shwartz-Ziv and Tishby (2017) - considers representations across 10^4 epochs of training on a dataset. The LLMs studied here largely complete only a single epoch over their training data, without seeing the dataset more than once. As a result of this and the overall complexity of language modelling as a task “zero error on ERM” does not occur. However we do think this is an important connection to be able to make with our work. As a result we have analysed the loss of each pretraining checkpoint with respect to c4, what we see is that the compression phase corresponds to a step-change in the loss value. While the loss doesn't reach zero it does saturate at the point at which representations begin to approach the bound. We believe this is directly related to the zero ERM phenomena in the original IB paper, and is a nice addition to the work here, thank you for the suggestion. This analysis can now be found in figure 1 and Appendix F.

---

> ### Author Response · Authors · 2025-11-26
>
> ## Weaknesses
>
> We group together weaknesses 1, 2, and 3 which address similar points.
>
> Regarding the difference between the facets in figure 1, and more generally the clarity of the transition between phases. The information bottleneck theory of deep learning only asserts that there is a fitting phase, followed by a compression phase where models approach the bound - this switch between phases results in the ‘knee’ in the trajectory through the information plane. The diagram on the left of figure 1 in our original submission was created by us in order to offer a hypothetical trajectory for what that process could look like, however it is only one example of what a valid IB trajectory could look like. Shwartz-Ziv and Tishby 2017 show how the compression phase may decrease both I(X;Z) and I(Y;Z) depending on the training data (figure 3 in their paper which we have reproduced in appendix H, along with substantial discussion).
>
> We believe our illustration on the left of figure 1 was misleading - it was intended to express that phase 1 expands the representation space, and phase 2 approaches the bound. Instead it expressed that there was a single valid trajectory. We have replaced this visual with a visual which shows representations distance to the bound over training, compared with the model’s loss. This shows there’s an initial phase of minimising error, followed by representations approaching the bound - the key hallmark of the compression phase. In line with a point you raise in your questions this also allows us to relate when the transition to compression happens to previous work looking at error saturation on the training data.
>
> Weakness 4
>
> We note in equation 5 that the entropies here are normalised to lie between 0 and 1. As a result the axes in all information plane figures are proportions, reflecting shift in the proportion of total possible mutual information. This means the change from 0.22 to 0.2 is a shift from 22% to 20% of the total possible mutual information - rather than a shift of only 0.02 bits. We also note that token information changes the most, with increasingly granular contextualisations having smaller shifts over the course of training (visible in figure 2).
>
> Weakness 5
>
> In response to this we have rerun the pre-training analysis of the OLMo 1b, 7b and 32b models with true mutual information - taking an expectation over label-level mutual informations rather than a mean. These results are now influenced by the token probabilities in the c4 data used to compute the estimates. These timecourses are included in appendix G.3 with these results following the pattern of the previous results that use a mean. We do agree that this was a worthwhile additional experiment, in order to confirm the validity of the core results.

---

### Official Review · Reviewer_jNtt · 2025-10-31

**Soundness:** 3
**Presentation:** 2
**Contribution:** 3
**Rating:** 6
**Confidence:** 3

**Summary:**

The paper proposes a method to frame pre-training as lossy compression, by leveraging the representational space, i.e. the learned representations with information-theory methods. They use entropy and mutual information between input, hidden representations and output, using a scalable soft-entropy estimator, to compute complexity (compression) and expressivity (task information). The authors  demonstrate that during training LLMs, there are two phases under the Information Bottleneck theory perspective: an initial expansion of expressivity followed by a compression phase. They propose that models approaching the optimal compression bound tend to perform better on downstream tasks. The approach is novel and well-motivated, and the findings regarding compression during training are a valuable contribution.

**Strengths:**

- **S1** While the soft-entropy estimator is not a contribution of this paper itself, it seems to be the first to explain and implement the approach from Conklin (2025) to compute entropy and mutual information within LLMs to distinguish between complexity (compression) and expressivity (task-relevant information).
- **S2** The use of such metrics to track the behaviour of LLMs during pretraining is, to the best of my knowledge, novel. While the results for smaller models are inconclusive, the results for olmo2 7B and 32B during training align with the Information Bottleneck theory and paired with the results at the end of training for a vast array of models constitutes a welcomed new perspective on LLM training.
- **S3** The paper is clear in its motivation for why information-theoretic analysis is needed for understanding LLMs, and it provides a nice overview on representation learning in the context of compression.
- **S4** The proposed methods to assess expressivity and compression during training are likely to spark future work exploring these further.

**Weaknesses:**

- **W1. Attribution and Contextualization of Contributions**
   - One of the main contributions of the paper is the implementation of the theoretical framework from Conklin (2025) to estimate soft-entropy. You should better contextualize this, not to diminish your own contributions, but to clarify that much of Section 3.1 is a reinterpretation of prior work.
   - Then, emphasize that this paper is the first to leverage the soft-entropy estimator in the context of LLMs.

- **W2. Inconsistent Level of Detail**
  - The level of detail varies across the main body, which may be challenging for readers who are not experts in compression.
  - For example, Section 3.2 glosses over the n-gram backoff mechanisms and provides little explanation of I(Z;Y) in the main text.
  - Readers would not know that I(Z;Y) is better explained in Appendix E.2 from reading the main body
  - In that vein, the paper seems to expect readers to assume that the mechanisms for I(Z;X) and I(Z;Y) are identical (see last sentences of E.2), but this is only made explicit in the appendix.
  - Meanwhile, the Introduction and Background sections are lengthy (up to page 5), and could be condensed to make room for more technical details that are currently omitted or relegated to the appendix.

- **W3. Section 4.2 Results and Interpretation**
  - The results in Section 4.2 are not fully convincing.
  - While it is good to see other use cases for the mutual information values, this section feels lackluster in achieving its goal.
  - It seems almost granted that I(Y;Z) will correlate with downstream performance.
  - The paper argues that the ratio with compression (expressivity/compression) is important, suggesting that two models with the same expressivity may not generalize equally and that compression may be the key difference.
  - However, the paper does not fully explore whether it is solely I(Y;Z) that is driving the correlation, and the claim of "optimal compression" may be more semantic than substantive without further analysis.

- **W4. Practical implications**
  - The paper focuses on empirical results, reporting results for a vast array of models, and checkpoints for the case of Olmo/Pythia.
  - However, not that much emphasis is given to practical implications (beyond what is discussed in W3).
  - This seems a missed opportunity. For instance, could these findings be used to predict early stopping?
  - What about when to expect the change between phase1 and phase2? Are there any implications related to that in terms of performance? (i.e. does the point in which a model start compressing, predict performance?) It does seem that larger models take proportionally more steps in phase 1.
  - Going back to W2, some content could have been condensed to leave place for more practical implications.

**Questions:**

- Have you analyzed whether the correlation with downstream performance is primarily driven by I(Y;Z) alone, or does the expressivity/compression ratio provide additional predictive power? Could you provide results or discussion to clarify this point?
- Are there other practical implications of your findings for LLM training? You could place here the questions from W4.

---

> ### Author Response · Authors · 2025-11-26
>
> We would like to thank you for the thorough and considered review. You raise a number of points about how details in the paper could be made more clear, which we agree with and have implemented. We briefly summarise changes made in response to your feedback before itemising our response to each weakness and question raised below.
>
> - In response to W3 We have added correlation results showing neither expressivity nor complexity alone predict performance (figure 4).
> - In response to W2 We have better clarified the estimation and backoff procedures in section 3.2
> - In response to W4 We have added potential applied use cases of our approach to section 4.2, to compliment the array of empirical results
> - In response to W1 we have further clarified that the soft entropy estimator is not original to our work, but that this represents a novel application in section 3.1.
>
> ## Weaknesses and Questions
>
> Weakness 1. Clarifying Relation to Previous work.
>
> - We have updated section 3.1 to better clarify the contributions of this work and previous work. You are correct that we do not introduce the soft entropy estimator, but are the first to apply it to study rate-distortion theory in LLMs.
>
> Weakness 2. Clarifying the Backoff and Labelling Procedure in the Main Text.
>
> - We agree that the exact details of the backoff and labelling procedure are critical for readers to have a clear understanding of what the estimates reflect. At the same time we include the lengthy background and related work because a core goal of our work is to place LLMs in appropriate theoretical context for learning across the sciences. As a result we have opted not to reduce the size of the background but to use the additional page included in the rebuttal version to move information about the labelling procedure from the appendix to the main body, and make the backoff procedure more accessible to a non-expert audience. We have made clear that the backoff procedure is symmetric of I(X;Z) and I(Y;Z) (starting at line 282). We also make clear that I(Z;Y) is based on approximations of the entire trailing context, and point readers to appendices E.1 and E.2 for further detail.
>
> Question 1/Weakness 3 | Does Expressivity Alone Predict Performance?
>
> - This is an excellent point that we would love to clarify. We have analysed whether or not expressivity I(y;z) alone is a significant predictor of downstream performance, and it is not. Neither complexity, nor expressivity alone has a significant relationship with performance in our experiments - it is instead their proportion which proves significant. This is the basis for our arguing that distance to the IB bound (what we term optimality) is what matters, rather than either quantity in isolation. In the original submission we included only the results regarding complexity, optimality, and preference information. We have updated these results to include the null result for relating expressivity and downstream performance.
>
> - Intuitively this relates to results showing that measures like validation loss - a kind of distortion measure in its own right - do not reliably predict downstream task accuracy in LLMs. Our results help understand why this may be; as discussed in our literature review in section 2.1 a large body of work makes the case that the best generalising solution is the most compressed compatible with the data. The solution with the lowest distortion would be a lossless encoding, which would likely struggle to generalise to novel tasks. As a result a distortion measure like expressivity, or loss, may be useful for predicting in-distribution performance but in more challenging generalisation cases a measure of how optimally compressed a model is - a compound notion of distortion and rate - may be needed. This is deeply related to the bias-variance tradeoff (mentioned at line 120) which articulates how more complex models may provide better fit to the in-distribution but struggle to generalise.

---

> ### Author Response · Authors · 2025-11-26
>
> ### Continued
>
> Question 2 / Weakness 4 | Discussing Potential Applications of our Methods.
>
> Again, this is a point well taken. We have strong ambitions that the framework and methods introduced  here have a variety of potential applications for guiding decision making in pre-training and post-training. However, given that our paper is work on interpretability, and we do not actually run experiments pre-training or post-training models guided by our methods we wrote the results and discussion section conservatively; stating the empirical results we knew we had and were significant, while avoiding speculative claims about what they could be used for.
>
> But we agree that some of this speculation is useful to place the work in context for the reader, allowing them to think about how the methods here relate to the practicalities of training. We’ve used the additional page for the rebuttal to add some discussion of the two potential applications we think are the most defensible to the results section:
>
> - Stopping-Criterion: As you note a natural application is for determining when to stop pre or post training. A pretraining stopping criterion could be when further training no longer moves representations closer to the bound. For post-training, when further training no longer increases the proportional amount of preference information.
>
> - Model-Selection Criterion: when selecting between model candidates, modellers currently tend to evaluate each candidate which requires inference across benchmarks and proves computationally expensive. Instead model selection could be based on which candidate is closest to the bound, or which has the highest relative preference information. Given the entropy estimate requires only a forward pass (with teacher forcing rather than full inference) this could dramatically reduce the cost of evaluating candidates, and avoid classic issues with benchmarks like dataset contamination.

---

### Official Review · Reviewer_WLPL · 2025-11-01

**Soundness:** 3
**Presentation:** 2
**Contribution:** 2
**Rating:** 2
**Confidence:** 3

**Summary:**

The paper investigates LLM training through the lens of Information Bottleneck (IB) theory, framing learning as a process of lossy compression of input information. It introduces an information-plane analysis that tracks the mutual information between model representations, inputs, and outputs during training, revealing a two-phase trajectory: first, expansion, then compression of representations. The authors develop a soft-entropy-based estimator to compute mutual information efficiently at LLM scale, enabling comparisons across model families and token contexts. Their empirical results show that models approaching optimal compression achieve higher accuracy on downstream benchmarks such as MMLU-Pro, and that post-training fine-tuning primarily increases human preference information rather than token-level complexity.

**Strengths:**

1. The paper presents a clear, interpretable information-theoretic framework for understanding representational dynamics of LLMs.

2. This paper evaluates multiple open-weight models, scaling classical IB analyses to billions of parameters.

3. This paper provides a readable exposition with intuitive information-plane plots.

**Weaknesses:**

1. The core argument that LLM training is a process of lossy compression approximating the Information Bottleneck largely restates conclusions from prior work, such as Language Modeling is Compression [1] and Norm of Word Embedding Encodes Information Gain [2]. The paper extends scale but does not introduce a new theoretical insight or training principle.

2. No comparison with established compression baselines. The study fails to relate its “compression optimality” metric to practical model compression methods like pruning [5, 6], quantization [7, 8], or low-rank adaptation [9, 10]. Without such baselines, it is unclear whether the proposed metric offers predictive or practical value for compression performance.

3. The observed correlation between compression optimality and MMLU-Pro accuracy does not imply causality. Unlike controlled studies of feature emergence or SGD-noise-driven IB dynamics [3, 4], the paper lacks interventions demonstrating that manipulating compression affects downstream accuracy.

4. The entropy estimation approach is based on a soft-entropy approximation but is not compared against common mutual-information estimators such as MINE or CLUB. Without quantitative validation, the robustness of the reported information-plane trajectories remains uncertain.

References

[1] Delétang et al., Language Modeling Is Compression, NeurIPS, 2023.

[2] Oyama et al., Norm of Word Embedding Encodes Information Gain, EMNLP, 2023.

[3] Tishby and Zaslavsky, Deep Learning and the Information Bottleneck Principle, ITW, 2015.

[4] Saxe et al., On the Information Bottleneck Theory of Deep Learning, J. Stat. Mech., 2019.

[5] Frantar and Alistarh, SparseGPT: Massive Language Models Can Be Accurately Pruned in One-Shot, arXiv, 2023.

[6] Ma et al., LLM-Pruner: On-the-Fly Structured Pruning for Large Language Models, NeurIPS, 2023.

[7] Frantar et al., GPTQ: Accurate Post-Training Quantization for Generative Pre-Trained Transformers, arXiv, 2022.

[8] Lin et al., AWQ: Activation-Aware Weight Quantization for LLMs, MLSys, 2024 (preprint 2023).

[9] Hu et al., LoRA: Low-Rank Adaptation of Large Language Models, ICLR, 2022.

[10] Zhang et al., AdaLoRA: Adaptive Budget Allocation for Parameter-Efficient Fine-Tuning, ICLR, 2023.

**Questions:**

Under matched compute/bit-budget/latency, how does the proposed “compression optimality” metric predict retained accuracy compared with established compression methods?

How robust are the information-plane trajectories to the choice of MI/entropy estimator? Can you benchmark your soft-entropy approach against common alternatives (e.g., MINE, CLUB)?

What concrete theoretical insight or training principle, beyond prior claims that “language modeling is compression” and IB-style dynamic, does this paper introduce?

---

> ### Author Response · Authors · 2025-11-26
>
> We would like to flag to the Area Chairs that we believe this review was written by an LLM. It conflates substantially different concepts which are only conceptually related to compression (weakness 2, question 1), and asserts that our work lacks novelty while referring to previous work whose claims do not overlap with our own, and which are already referenced in our submission.
>
> Additionally AI detection by Pangram has flagged this review as fully AI generated: https://iclr.pangram.com/reviews?submission_number=21536
>
> While we have written a rebuttal to this review, we believe this reviewer has not engaged in good faith.
>
> Weakness 1 | Lack of Novelty; Our Paper Restates Conclusions from Previous Work.
>
> Critically, the citations you provide do not actually state conclusions from our work. For [1], if you read section 2.1 of our submission we explicitly mention this paper and its relation to our own work at lines 122-125. Deletang et al. (2023) study the output distributions from LLama 2 and Chinchilla at the end of training. By feeding raw bit strings into the model they use the output distributions to derive an arithmetic encoder and evaluate how well the model can compress the raw bit strings in comparison to lossless compression algorithms like gzip. It is hopefully clear how this is distinct from the work here; we study the actual model-internal representations (rather than output probabilities), based on input text (rather than a raw bit-string encoding), over the course of training (rather than just taking a trained model), looking at how representations evolve over training in terms of lossy compression (Deletang explicitly states “we advocate for using (lossless) compression to study foundation models” which is a different area of information theory from lossy compression). As our paper notes “While some work has studied whether or not LLMs can match lossless compression algorithms in-context (e.g. Delétang et al., 2023), this is distinct from giving an account of LLM training itself as a process of lossy compression – the object of study here.”
>
> Regarding the second citation provided here Oyama et al. (2023)  give an account of how the norm of an embedding vector relates to the probability of the token it corresponds to in the training corpus. While this is an interesting result (and one we mention in the appendix at line 1142, given the estimator we use explicitly does not consider the norm information which is the study of Oyama’s work), Oyama et al. (2023) does not discuss compression, or lossy compression, or formalise pretraining as a process of lossy compression. It relates to our work in that it uses information theoretic principles to look at a model’s embeddings, but to say that a weakness of our work is that it merely restates conclusions from Oyama et al. (2023) would seem to indicate a fundamental lack of understanding of both our paper and theirs.
>
> Question 1 & Weakness 2 | Did We Compare Our Methods with Pruning, Quantisation, or LoRa?
>
> We thank the reviewer for this question. We believe it highlights an important distinction between engineering-driven compression techniques (e.g., quantization and pruning) which aim to reduce the memory footprint of a model, and the information-theoretic compression studied in our work. As we note in section 2 our approach to compression follows in the information theoretic tradition, and we study how representations in a model represent information present in the training data. In this context compression occurs when the information “rate” - the amount of information preserved about the input - is reduced. In this paper the rate is referred to as “complexity” and quantified as the mutual information between the input and a model’s embeddings I(X;Z). Compression occurs when I(X;Z) decreases which involves no change to the parameter count (pruning), or the float precision (quantisation), or use of a low rank approximation to further train the model (LoRA). As a result the approaches you mention which reduce the memory footprint of a model are not baselines for our study, which looks at compression in models’ representational structure rather than memory footprint. While there may be interesting research questions about how representational compression relates to approaches like pruning, pruning is not a ‘baseline’ for computing an information rate.

---

> > ### Author Response · Authors · 2025-11-26
> >
> > ### *Continued*
> >
> >
> > Question 2 & Weakness 4 | Did We Compare Different Entropy Estimators?
> >
> > We have not analysed pretraining trajectories with alternate estimators like MINE. The pretraining analyses presented in the paper are expensive, requiring computing embeddings with respect to 10,000 samples from the c4 dataset across 100s checkpoints for each of the models studied. An estimator like MINE is a gradient based approach which would require training until convergence at each checkpoint in addition to the existing computational overhead of embedding 10,000 samples from c4. As we note part of our reason for selecting the soft entropy estimator is its scalability which makes these analyses tractable to compute across the volume of checkpoints we study. Conklin (2025) introduces the soft entropy estimator, comparing it with existing estimators on reference distributions to show similar convergence behaviour. Our paper is an analysis of LLM pretraining rather than a paper that introduces a novel entropy estimator. We leave broader discussion of entropy estimation for continuous spaces, and a broad comparison of existing methods to future work in that domain.
> >
> > Question 3 | What Concrete Insights aside from “language modeling is compression” Does This Paper Introduce?
> >
> > While we believe our work asserts other “concrete” insights we would first like to note that this question trivialises the results presented here. While previous work has looked at deep-learning models from an IB perspective, these analyses have been at substantially smaller scale. Just in the papers you cite [3] is a purely theoretical workshop paper with no empirical results, and [4] studies feed-forward networks on MNIST. Additionally our related work is 2 pages in length and discusses both of these papers (along with many others) and how they relate to our work. Formalising the process of pretraining as a rate-distortion problem at 32 Billion parameter scale is a substantive contribution in its own right.
> >
> > Further to this point - as noted above in our response to weakness 2 - your citations [1] and [2] also do not include claims that overlap with our own. While papers have looked at if LLMs can generate compressed sequences [1] this does not study model training or representations. Your review generally questions the novelty of our findings by offering previous work, all of which appears in our paper’s bibliography, all of which predominantly overlaps with our work not in results or findings, but in titles and keywords.
> >
> > With that being said, in addition to being the first paper to show LLM pretraining follows the trajectory predicted by the Information Bottleneck, our work makes a number of contributions. We show how size conditions representational structures with smaller models struggling to achieve meaningful compression (line 413). We show how models predominantly represent local contextual information (line 370). We relate optimality of a compression to downstream performance (line 477). We also quantify the amount of preference information present in a model, showing how that relates to performance (line 487), and in the appendix show that post-training increases the amount of preference information in a model (line 900). In terms of offering a concrete training principle, this suggests better models may be more compressed and have proportionally more preference information. Future work could try to operationalise these as stopping criterions for pre-training and post-training or as a model selection criterion. We have added discussion of these potential applications of our methods to section 4.2 (line 498) but leave examination of this possibility to papers that try to improve model training rather than study representational structure.

---

### Official Review · Reviewer_niuc · 2025-11-01

**Soundness:** 1
**Presentation:** 1
**Contribution:** 1
**Rating:** 0
**Confidence:** 5

**Summary:**

This work frames large language models as lossy compressors that retain only information essential to their training goals. The paper tries to show that that pre-training drives models toward Information Bottleneck optimality, and differences in compression across model families reflect data and training choices. Very limited and insufficient empirical studies are provided to support the claim.

**Strengths:**

None noted.  See explanation in the weaknesses.

**Weaknesses:**

1. The authors ignore that the understanding LLMs from the perspective of information bottleneck (IB) has been explored in the literature, such as [1-3]. These existing works understanding the compression effect of variants of LLMs by IB, and none of these works are cited and discussed in this paper. The main claimed contribution of this paper, the compression effect of LLMs from an IB persecuted, is not new and already discussed in these existing works.

2. While this paper was submitted as a "learning theory" paper, no formal theoretical results are presented and proved, far below the standard of major machine learning venues including ICLR.

3. Again, this paper lacks a key formal definitions to support its main claim, While "optimal compression" of LLMs are mentioned across this paper for many times, there no formal result about when "optimal compression" can happen (that is, minimum of Eq. (1)), and there is no result about why training LLMs can achieve such "optimal compression".

4. The formatting and presentations are chaotic, far below the bar of the a major ML venue. A lot of sentences are ungrammatical, and there are a lot of formatting issues (large space between equations or paragraphs).

5. This paper also lacks sufficient empirical justification: experiments on large-scale LLMs on standard benchmark should be used to support the claims, such as LLaMA-4 and Qwen-3.


[1] Protecting Your LLMs with Information Bottleneck. NeurIPS 2024.

[2] An Information Bottleneck Perspective for Effective Noise Filtering on Retrieval-Augmented Generation. ACL 2024.

[3] QUITO-X: A New Perspective on Context Compression from the Information Bottleneck Theory. EMNLP 2025.

**Questions:**

See weaknesses.

---

### Author Response · Authors · 2025-11-26
**Overall Response to Reviewers**

Reviewers,

We would like to thank all of you for taking the time to review and engage with our work. Our delay in replying to reviews has been due to a substantial number of additional experiments which we have run in order to address insights and concerns raised as part of the review process - this required analysing over 800 model checkpoints. We have replied to reviewers individually but summarise here some of the supplementary experiments and other changes that have been added to the paper. We thank you for the feedback, and we believe the additions and changes made in response have made our work stronger.

- In response to asking if expressivity alone predicts performance we have added correlation results showing neither expressivity nor complexity alone relate significantly to MMLU Pro performance (figure 4). Only their ratio - our distance to bound measure - has a significant relationship with performance.
- We have added an analysis relating a model’s loss on next token prediction to our distance to bound measure in figure 1 of the main paper and Appendix F. We show the OLMo2 7b model enters the compression regime approaching the bound on optimal compression as the training loss saturates. This is an analogue to the previous work relating “zero error” to compression. (this required computing the entropy analysis for 106 additional 7B model checkpoints)
- We have added a robustness analysis (Appendix G) which shows the core two-phase pattern of training is robust to the number of points used in W as well as the data distribution. This required re-running the pretraining timecourse of the 7B model with half the number of points, and with data from C4, Tulu, and MMLU (this required computing the entropy analysis for 424 additional 7B model checkpoints)
In response to concerns about our taking mean label-level mutual informations rather than an expectation, we have added the pretraining analysis for the OLMo2 1B, 7B, and 32B models computed with an expectation to appendix G.3 showing this does not substantially change the pretraining trajectory through the information plane. (This required computing the analysis for 283 different model checkpoints of various sizes)
- We have better clarified the estimation and backoff procedures in section 3.2
- We have added potential applied use cases of our approach to section 4.2, to compliment the array of empirical results

We will continue to check back here, and are happy to answer any further questions or clarifications. Thank you again for the time taken to engage with our work.

---

### Meta-Review · Area_Chair_Q1vT · 2026-01-07

**Summary:**

The reviewers’ main concerns focus on clarifying how this work differs from prior literature—both on applying the Information Bottleneck perspective to LLMs and on using entropy/mutual-information estimators—along with questions about the practical utility of the paper’s observations. The authors attempt to address these points, and the work appears to be among the first to conduct this kind of analysis at relatively large scale in the LLM setting.

**Reviewer Concerns:**

The authors clarified their contributions more clearly, emphasizing that the novelty lies in the LLM-scale application and the Information Bottleneck / rate–distortion analysis rather than the soft-entropy estimator itself. They also added correlation analyses showing that neither expressivity nor complexity alone reliably predicts downstream performance, while the expressivity-to-compression ratio (or distance-to-bound) is the meaningful signal. Overall, I think this provides a useful lens for the community to understand LLM training dynamics. However, the paper is still less convincing on practical utility: the rebuttal discusses potential applications, but it does not fully demonstrate how the analysis and theory translate into actionable training or model-selection procedures.

**Reviewer Scores:**

The reviewer niuc and WLPL look significantly negative to the paper and the other two reviewers are positive. The authors also addressed the reviewers concerns well.

---

### Decision · Program_Chairs · 2026-01-26

Accept (Poster)